

# A new uncertainty estimation technique for multiple datasets and its application to various precipitation products

Xudong Zhou[1,2], Jan Polcher[2], Tao Yang[1], and Ching-Sheng Huang[1]

[1]State Key Laboratory of Hydrology-Water Resources and Hydraulic Engineering, Center for Global Change and Water Cycle, Hohai University, Nanjing 210098, China
[2]Laboratoire Météorologie Dynamique du CNRS, IPSL, CNRS, Paris, F 91128, France

**Abstract.**

The uncertainty among climatological datasets can be characterized as the variance in space and time between various estimates of the same quantity. However, some of the current uncertainty estimates only evaluate variations in one single dimension (time or space) due to the limitation of estimation methodology as averaging variation is necessary for the other dimension. The influence on the uncertainty assessment of the ignorance of variations in one dimension is not well studied. This study introduces a new three-dimensional variance partitioning approach which avoids the averaging and provides an new uncertainty estimation ($U_e$) technique with consideration of both temporal and spatial variations. Comparisons of $U_e$ to classic uncertainty estimations show that the classic metrics underestimate the uncertainty because of the averaging of variation in the time or space dimension, and $U_e$ is around 20% higher than classic estimations. The deviation between the new and classic metrics is higher for regions with strong spatial heterogeneity and where the spatial and temporal variations significantly differ. Decomposing of the new metric demonstrates that $U_e$ is a comprehensive assessment of model uncertainty which has been included the model variations identified by the classic metrics. Multiple precipitation products of different types (gauge-based, merged products and GCMs) are used to better explain and understand the peculiarity of the new methodology. The new uncertainty estimation technique is flexible in its structure and particularly suitable for a comprehensive assessment of multiple datasets over a large regions within any given period.

## 1 Introduction

With the technical development for monitoring the natural climate variables and the increasing knowledge of the physical mechanisms in the climate system, many institutes have the ability to provide different kinds of climate datasets. Taken the precipitation, which is the dominant variable in the land water cycle, as an example, there are point measurements as GHCN-D (global historical climatology network-daily, Menne et al., 2012) , grid products based on gauge measurements and interpola-



tion (e.g., CRU, Harris et al., 2014) , products derived from remote sensing (e.g., the Tropical Rainfall Measuring Mission - TRMM), reanalysis datasets (e.g., NCEP) and those estimates from models (e.g., GCMs). These products are developed using different original data, different technologies or different model settings for various purposes (Phillips and Gleckler, 2006; Tapiador et al., 2012; Beck et al., 2017; Sun et al., 2018). Therefore, there are differences (including systemic bias and un-

certainty) among the various products, and the uncertainty can be regarded as the deviation around what is believed to be the truth.

Although many studies have attempted to understand the causes of the uncertainties in different products, the uncertainties are very difficult to be removed from datasets. Thus, using ensembles consisting of multiple datasets to generate a weighed average has become popular in the climate-related researches. For example, the IPCC uses 42 CMIP5 (Coupled Model Inter-

comparison Project Phase 5) models to show the historical temperature change and 39 CMIP5 models to average the temperature projection in future RCP 8.5 scenario (Figure SPM.7 in IPCC, 2013b). Schewe et al. (2014) use nine global hydrological models to evaluate the global water scarcity under climate change. GLDAS (Global Land Data Assimilation System) involves four different land surface models (Rodell et al., 2004) and GRACE (Gravity Recovery and Climate Experiment) provides estimations from three independent institutes (Landerer and Swenson, 2012). Using multiple datasets reduces the dependence

on a single dataset and decreases the risk of using a single dataset which might contain undiscovered uncertainties.

Extra uncertainty information has to be provided along with the ensemble means because the uncertainty influences the significance or the reliability of the ensemble result. In general, the uncertainty can be quantified as the range of maximum and minimum values (i.e., $V_{max} - V_{min}$), range of values at different quantiles (e.g., $V_{5\%} - V_{95\%}$), the consistency of models as the ratio of models following a certain pattern to the total number of models, the variation ($\sigma^2$) or the standard deviation

($\sigma$), which represents different characteristics of the multiple datasets. Among the uncertainty metrics, the standard deviation ($\sigma$) is the most used because it has the same magnitude as the original dataset; it avoids influence of extreme samples and it is less sensitive to the number of datasets used for the investigation. The ratio of the standard deviation ($\sigma$) to the mean value ($\mu$), which is called coefficient of variance ($CV$), representing the dispersion or spread of the distribution of various ensemble members (Everitt, 2013), is a unit-less value which also shows the degree of uncertainty.

Depending on the purpose of the evaluation, the uncertainty among datasets can be displayed over space to show the spatial heterogeneity of the consistency among multiple datasets. For example, the predicted future temperature increase has a higher significance in the northern high-latitudes among different models than in the middle-latitudes (Box TS.6 Figure 1 in IPCC, 2013a). The other typical implementation is to evaluate the evolution of the model uncertainty across time. In general, the uncertainty range decreases in the historical period over time because more observations are accessible in recent while the

uncertainty increases for future projections because the increasing spread of the model simulations (Figure SPM.7 in IPCC, 2013b). The increasing uncertainty range indicates the decreasing of consistency and increasing variations among various datasets.

The above metrics have been widely used as they show the temporal evolution or spatial distribution of the uncertainty easily. But the short-coming is obvious as we have to average the values in one of the dimensions (time or space, Figure 1)

when we use either of the assessments for specific purpose. For example, the averaging over a specific region (spatial mean) is





estimated at each time step before the temporal evolution of the model uncertainty can be obtained (red flowcharts in Figure 1). And the averaging over a certain period (temporal mean) is estimated at each grid cell before the spatial distribution of the model uncertainty can be obtained (blue flowcharts in Figure 1). While, the averaging in either dimension means a loss of the information, especially the data variability in that dimension. This may result in that the uncertainty among datasets not being

fully considered when estimating the uncertainties. In other words, either of the uncertainty estimates cannot represent the full differences among datasets. Therefore, the uncertainty among datasets can be underestimated and the similarity among them can be overestimated with these two procedures. However current studies have not paid attention to the ignorance of variation due to the averaging as well as its influence on the uncertainty assessment.

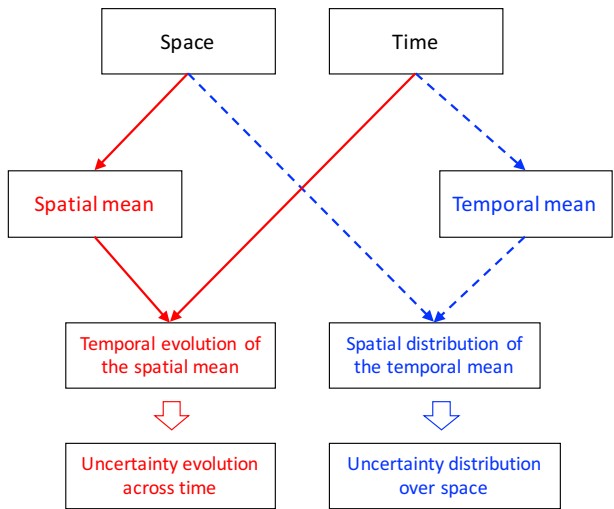

**Figure 1.** The two classic uncertainty assessments in the current researches as the temporal evolution of the model uncertainty (red) and the spatial distribution of the model uncertainty (blue). Either of the uncertainty estimates has to do the averaging in one of the dimensions in space or time, and it will lead to the loss of information in the corresponding dimension.

In this study, we aim to introduce a new technique for uncertainty estimation among multiple datasets. The new uncertainty

estimation technique avoids any averaging in time or space dimension thus all the information across the two dimensions can be maintained for the uncertainty assessment along with an ensemble of estimates. Multiple precipitation products are used to explain the peculiarity of the new methodology. In section 2, the detailed methodology of the three-dimensional variance partitioning approach is introduced. The characteristics of multiple precipitation datasets and estimations of the two classic uncertainty metrics are shown in section 3. The results of the new approaches for the precipitation products are discussed

in section 4. The differences between the new uncertainty estimation and the two classic metrics introduced previously are analyzed and the causes are discussed in section 5. The discussion and conclusions are followed in the end of this article.





## 2 Methodology and datasets

### 2.1 Mathematical Derivation

The multiple climatic dataset have to be organized in three dimensions of (1) **time** with a regular time interval (e.g. monthly or annual), (2) **space** with regular spatial units where all the grids are re-organized in a new dimension from the original latitude-longitude grids, (3) **ensemble** with different ensemble datasets regarded as the third dimension. Thus, the dataset array can be reformed as

$$\mathbf{Z} = [z_{ijk}] \tag{1}$$

with $i$-th time step ($i = 1, 2, \ldots, m$), $j$-th grid ($j = 1, 2, \ldots, n$), and $k$-th ensemble member or ensemble model ($k = 1, 2, \ldots, l$).

We define the three dimensions as time, space and ensemble dimension and the means for these three dimensions are called temporal mean, spatial mean and ensemble mean, respectively. The corresponding variances are named time variance, space variance and ensemble variance, respectively. The grand mean ($\mu$), grand variance ($\sigma^2$) across time, space and ensemble dimensions as well as the total sum of squares ($SST$) are defined as.

$$\mu = \sum_{i=1}^{m}\sum_{j=1}^{n}\sum_{k=1}^{l} z_{ijk}/(mnl) \tag{2}$$

$$\sigma^2 = \frac{SST}{mnl} \tag{3}$$

$$SST = \sum_{i=1}^{m}\sum_{j=1}^{n}\sum_{k=1}^{l} (z_{ijk} - \mu)^2 \tag{4}$$

The total variations is contributed by the variation in all dimensions. Thus, it should be reformulated as an express of variations in three dimensions. The derivation of the total squares can start from the third ensemble dimension. For a specific $k^{th}$ ensemble member, the grand mean is formulated as $\mu_{ts}[k] = \sum_{i=1}^{m}\sum_{j=1}^{n} z_{ijk}/(mn)$, leading to the total squares rewritten as

$$SST = \sum_{i=1}^{m}\sum_{j=1}^{n}\sum_{k=1}^{l} (z_{ijk} - \mu_{ts}[k] + \mu_{ts}[k] - \mu)^2 \tag{5}$$

and then expanded and rearranged as

$$SST = \sum_{i=1}^{m}\sum_{j=1}^{n}\sum_{k=1}^{l} (z_{ijk} - \mu_{ts}[k])^2$$
$$+ 2 \times \sum_{k=1}^{l} (\mu_{ts}[k] - \mu) \underbrace{\left[ \sum_{i=1}^{m}\sum_{j=1}^{n} (z_{ijk} - \mu_{ts}[k]) \right]}_{=0}$$
$$+ \underbrace{\left[ \sum_{i=1}^{m}\sum_{j=1}^{n} \right]}_{=mn} \sum_{k=1}^{l} (\mu_{ts}[k] - \mu)^2 \tag{6}$$





$$SST = \sum_{i=1}^{m}\sum_{j=1}^{n}\sum_{k=1}^{l}(z_{ijk} - \mu_{ts}[k])^2 + mn\sum_{k=1}^{l}(\mu_{ts}[k] - \mu)^2 \tag{7}$$

$$SST = mn\sum_{k=1}^{l}\sigma_{ts}^2[k] + mnl\sigma^2(\mu_{ts}) \tag{8}$$

Where $\sigma^2(\mu_{ts})$ is the variation of the grand mean for each member of the ensemble, and $\sigma_{ts}^2[k]$, the grand variance in space and time for ensemble member $k$, can be split using the mean of the spatial variation at each time step $\overline{\sigma_s^2[k,:]}$ and the variation of the spatial mean $\sigma^2(\mu_s[k,:])$, denoted as

$$\sigma_{ts}^2[k] = \overline{\sigma_s^2[k,:]} + \sigma^2(\mu_s[k,:]) \tag{9}$$

  The detailed derivation of Eq. (9) is shown in Eqs. (10) - (17). For a specific dataset $k$, the grand mean $\mu_{ts}[k]$ through
space-time scale is

$$\mu_{ts}[k] = \frac{1}{mn}\sum_{i=1}^{m}\sum_{j=1}^{n}z_{ijk} \tag{10}$$

The total squares for difference from the grand mean is

$$SST[k] = \sum_{i=1}^{m}\sum_{j=1}^{n}(z_{ijk} - \mu_{ts}[k])^2 \tag{11}$$

and the grand variance $\sigma_{ts}^2$ is

15 $$\sigma_{ts}^2[k] = \frac{1}{mn}\sum_{i=1}^{m}\sum_{j=1}^{n}(z_{ijk} - \mu_{ts}[k])^2 \tag{12}$$

If the derivation is started from the space dimension, Eq. (11) can be rewritten by incorporating the spatial mean of each time step $\mu_s[k,i] = \sum_{j=1}^{l}z_{ijk}/n$

$$SST[k] = \sum_{i=1}^{m}\sum_{j=1}^{n}(z_{ijk} - \mu_s[k,i] + \mu_s[k,i] - \mu_{ts}[k])^2 \tag{13}$$

It can be expanded and then rearranged as

$$
\begin{aligned}
SST[k] = & \sum_{i=1}^{m}\sum_{j=1}^{n}(Z_{ijk} - \mu_s[k,i])^2 \\
& + 2\times\sum_{i=1}^{m}(\mu_s[k,i] - \mu_{ts}[k])\times\underbrace{\left[\sum_{j=1}^{n}(Z_{ijk} - \mu_s[k,i])\right]}_{=0} \\
& + \underbrace{\left[\sum_{j=1}^{n}\right]}_{=n}\sum_{i=1}^{m}(\mu_s[k,i] - \mu_{ts}[k])^2
\end{aligned}
\tag{14}
$$




$$SST[k] = \sum_{i=1}^{m}\sum_{j=1}^{n}(Z_{ijk} - \mu_s[k,i])^2 + n\sum_{i=1}^{m}(\mu_s[k,i] - \mu_{ts}[k])^2 \tag{15}$$

$$SST[k] = n\sum_{i=1}^{m}\sigma_s^2[k,i] + nm\sigma^2(\mu_s[k,:])$$
$$= nm\overline{\sigma_s^2[k,:]} + mn\sigma^2(\mu_s[k,:]) \tag{16}$$

$$\sigma_{ts}^2[k] = \frac{SST[k]}{mn} = \overline{\sigma_s^2[k,:]} + \sigma^2(\mu_s[k,:]) \tag{17}$$

Here $\overline{\sigma_s^2[k,:]}$ is the mean of the spatial variation at each time step and $\sigma^2(\mu_s[k,:])$ is the variation of the spatial mean. Or, the grand variance can be split using the average of the temporal variation from all regions $\overline{\sigma_t^2[:,k]}$ and the space variation of the temporal mean $\sigma^2(\mu_t[:,k])$ if we started from the time dimension:

$$\sigma_{ts}^2[k] = \overline{\sigma_t^2[:,k]} + \sigma^2(\mu_t[:,k]) \tag{18}$$

With Eq. (9) and Eq. (18), we can have

$$\sigma_{ts}^2[k] = \frac{1}{2}\left\{[\sigma^2(\mu_t[:,k]) + \overline{\sigma_s^2[k,:]}] + [\sigma^2(\mu_s[k,:]) + \overline{\sigma_t^2[:,k]}]\right\} \tag{19}$$

Substituting Eq. (19) into Eq. (8) results in

$$SST = \frac{mn}{2}\sum_{k=1}^{l}[\sigma^2(\mu_t[:,k]) + \overline{\sigma_s^2[k,:]}]$$
$$+ \frac{mn}{2}\sum_{k=1}^{l}[\sigma^2(\mu_s[k,:]) + \overline{\sigma_t^2[:,k]}] + mnl\sigma^2(\mu_{ts}) \tag{20}$$

The first term on the right-hand side of Eq. (20) can be transformed to:

$$\frac{mn}{2}\sum_{k=1}^{l}[\sigma^2(\mu_t[:,k]) + \overline{\sigma_s^2[k,:]}] = mnl\left[\frac{\overline{\sigma_{s\_t}^2} + \overline{\sigma_s^2}}{2}\right] \tag{21}$$

where $\overline{\sigma_{s\_t}^2}$ is the mean of space variation of the temporal mean across each ensemble member, $\overline{\sigma_s^2}$ represents the grand mean of $\sigma_s^2$, which is the grand variance across time and ensemble dimensions. Then Eq. (20) becomes:

$$SST = mnl\left[\frac{\overline{\sigma_{s\_t}^2} + \overline{\sigma_s^2}}{2}\right] + mnl\left[\frac{\overline{\sigma_{t\_s}^2} + \overline{\sigma_t^2}}{2}\right] + mnl\sigma_e^2(\mu_{ts}) \tag{22}$$

where $\overline{\sigma_{t\_s}^2}$ is the mean of time variation of the spatial mean across ensembles, $\overline{\sigma_t^2}$ represents the grand mean of $\sigma_t^2$, the grand variance across space and ensemble dimensions. $\sigma_e^2(\mu_{ts})$ represents the variation of the spatial-temporal means ($\mu_{ts}$). Similarly, the derivation can start from any of the other two dimensions. And the $SST$ derived from time and space dimensions are formulated, respectively, as

$$SST = mnl\left[\frac{\overline{\sigma_{s\_e}^2} + \overline{\sigma_s^2}}{2}\right] + mnl\left[\frac{\overline{\sigma_{e\_s}^2} + \overline{\sigma_e^2}}{2}\right] + mnl\sigma_t^2(\mu_{se}) \tag{23}$$


$$SST = mnl\left[\frac{\overline{\sigma_{e\_t}^2} + \overline{\sigma_e^2}}{2}\right] + mnl\left[\frac{\overline{\sigma_{t\_e}^2} + \overline{\sigma_t^2}}{2}\right] + mnl\sigma_s^2(\mu_{et}) \tag{24}$$

Where each variable is defined in the Appendix A. Averaging these three expressions of $SST$ defined in Eqs. (22) - (24) leads to

$$
\begin{aligned}
5 \quad SST = \quad & \frac{mnl}{3}\left[\frac{\overline{\sigma_{t\_s}^2} + \overline{\sigma_{t\_e}^2}}{2} + \overline{\sigma_t^2} + \sigma_t^2(\mu_{se})\right] \\
& + \frac{mnl}{3}\left[\frac{\overline{\sigma_{s\_t}^2} + \overline{\sigma_{s\_e}^2}}{2} + \overline{\sigma_s^2} + \sigma_s^2(\mu_{et})\right] \\
& + \frac{mnl}{3}\left[\frac{\overline{\sigma_{e\_t}^2} + \overline{\sigma_{e\_s}^2}}{2} + \overline{\sigma_e^2} + \sigma_e^2(\mu_{ts})\right]
\end{aligned}
\tag{25}
$$

With the total degree of freedom $(m \times n \times l)$, the grand variance is expressed as

$$
\begin{aligned}
\sigma^2 = \quad & \underbrace{\frac{1}{3}\left[\frac{\overline{\sigma_{t\_s}^2} + \overline{\sigma_{t\_e}^2}}{2} + \overline{\sigma_t^2} + \sigma_t^2(\mu_{se})\right]}_{V_t} \\[2mm]
10 \quad & + \underbrace{\frac{1}{3}\left[\frac{\overline{\sigma_{s\_t}^2} + \overline{\sigma_{s\_e}^2}}{2} + \overline{\sigma_s^2} + \sigma_s^2(\mu_{et})\right]}_{V_s} \\[2mm]
& + \underbrace{\frac{1}{3}\left[\frac{\overline{\sigma_{e\_t}^2} + \overline{\sigma_{e\_s}^2}}{2} + \overline{\sigma_e^2} + \sigma_e^2(\mu_{ts})\right]}_{V_e}
\end{aligned}
\tag{26}
$$

where $V_t$, $V_s$ and $V_e$ represent the time, space and ensemble variances, respectively. To facilitate the understanding of the partitioning results, an illustration of the present approach is shown in Figure 2.

Note that $V_e$ is only based on the combination of variation across the ensemble dimension. The four components are the variations of temporal and spatial values ($\overline{\sigma_e^2}$, zone B3), temporal mean ($\overline{\sigma_{e\_t}^2}$, zone C3), spatial mean ($\overline{\sigma_{e\_s}^2}$, zone C6) and the grand variance of the spatiotemporal mean for a single ensemble member ($\sigma_e^2(\mu_{ts})$, zone F3). Similarity, the other variances only rely on the variances in the corresponding dimension, which shows the independence in the three dimensions.

## 2.2 Metrics definition for model uncertainty

Since the temporal evolution or the spatial heterogeneity is natural in the climate variables and the purpose of this study is to evaluate the model uncertainty among datasets, we focus mainly on the variance in the ensemble dimension. The uncertainty among the ensemble member is normalized as the ratio of the square root of the ensemble variance ($V_e$) divided by the mean value of the datasets ($\mu$).

$$U_e = \sqrt{V_e}/\mu \tag{27}$$

For each basic spatial unit (grid cell in this study), we can estimate the long-term mean of the target variable for each dataset $\mu_t[j,k]$, $j = 1, ..., n$ represents the space unit, and $k = 1, ..., l$ represents the number of datasets. Then for each spatial unit, we





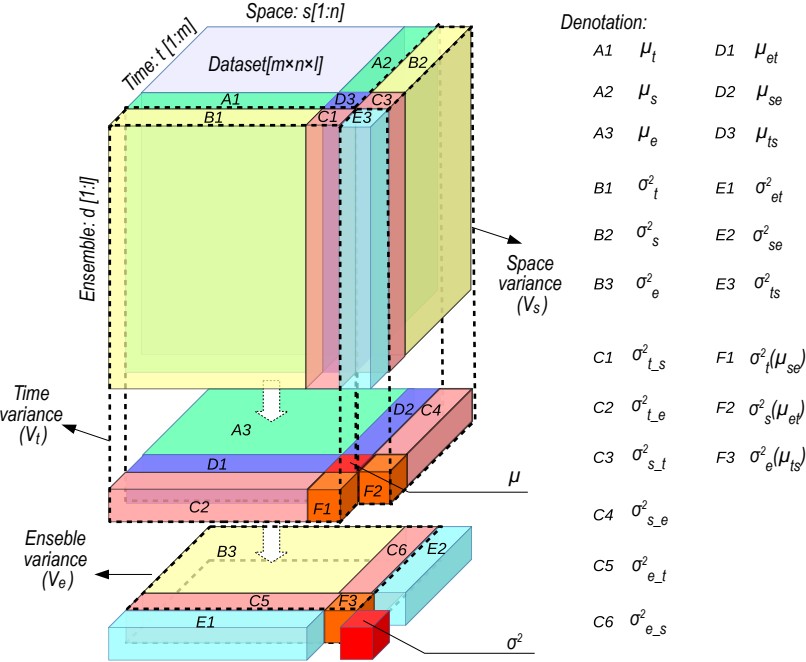

**Figure 2.** The illustration of the partitioning time-space-ensemble variance method. The original dataset is reorganized into three dimensions of time, space and ensemble. The denotations of the zones are listed to the right. The grand variance is defined as $\sigma^2$ and the grand mean as $\mu$. The subscripts $t$, $s$, and $e$ represent time, space and ensemble, respectively. Zone A ($\mu_i$) indicates the means of the $i$ dimension; zone B ($\sigma_i^2$) indicates the variation for $i$ dimension; zone C ($\sigma_{i\_j}^2$) indicates the variation across $i$ dimension of the means of $\mu_j$; zone D ($\mu_{ij}$) indicates the means across $i$ and $j$ dimensions; zone E ($\sigma_{ij}^2$) indicates the variation across $i$ and $j$ dimensions; zone F ($\sigma_i^2(\mu_{jk})$) indicates the variation across $i$ dimension of the means across $j$ and $k$ dimensions. The detailed definitions of these denotations can be found in Appendix A.

can estimate the ensemble variations across different datasets of the mean values as $\sigma^2(\mu_t[j,:])$ (expressed as $\sigma_{e\_t}^2[j]$ in this study). The spatial distribution of the $\sigma_{e\_t}^2$ shows the magnitude of model uncertainty over space and its root $\sigma_{e\_t}[j]$ is the model deviation at each space unit. The overall estimation of the model uncertainty over the entire region can be expressed as:

$$N.s.std = \sqrt{\overline{\sigma_{e\_t}^2}}/\mu = \frac{1}{\mu}\sqrt{\frac{1}{n}\sum_{j=1}^{n}\sigma_{e\_t}^2[j]} \tag{28}$$

5   $\sigma_{e\_t}^2[j]$ has different values for each spatial unit and the values for all the grid cells are averaged to obtain $\overline{\sigma_{e\_t}^2}$, which shows the general magnitude of the ensemble variation over space. The $N.s.std$ is normalized as the ratio of the square root of the mean of variations $\sqrt{\overline{\sigma_{e\_t}^2}}$ to the average value of all the datasets $\mu$.

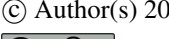



Similarly, the model uncertainty can also be expressed as the normalized as the ratio of the square root of the averaged ensemble variation at all time steps $\overline{\sigma^2_{e\_s}}$ to the entire means (Eq. 29).

$$N.t.std = \sqrt{\overline{\sigma^2_{e\_s}}}/\mu = \frac{1}{\mu}\sqrt{\frac{1}{m}\sum_{i=1}^{m}\sigma^2_{e\_s}[i]} \tag{29}$$

where the $\sigma^2_{e\_s}[i], i = 1,...,m$ is the ensemble variation of the spatial mean of each dataset across different datasets of the spatial
means of each products at each time unit $\mu_s[i,k], (i = 1,...,m, k = 1,...,l)$. It has different values at different time steps.

The two uncertainty estimates (Eqs. 28 and 29) correspond to the two classic metrics presented in the Introduction. And we will compare the $U_e$ with the two classic metrics ($N.t.std$ and $N.s.std$) to show their relations and differences.

## 2.3  Study area and data descriptions

China is large in its area, and different climate types are encountered in the mainland (Kottek et al., 2006). To facilitate the
comparisons and analyses that have spatial variations, ten different subregions are defined in Figure 3 as the (1) Songhua River Basin, (2) Liao River Basin, (3) Hai River Basin, (4) Yellow River Basin, (5) Huai River Basin, (6) Yangtze River Basin, (7) Southeast China, (8) South China, (9) Southwest China, (10) Northwest China. The entire Chinese mainland is numbered as the 11[st] region. Most of the regions are natural river basins, and this definition is more natural when considering water resources analysis than definitions using longitude-latitude grids or that are based on administrative regions.

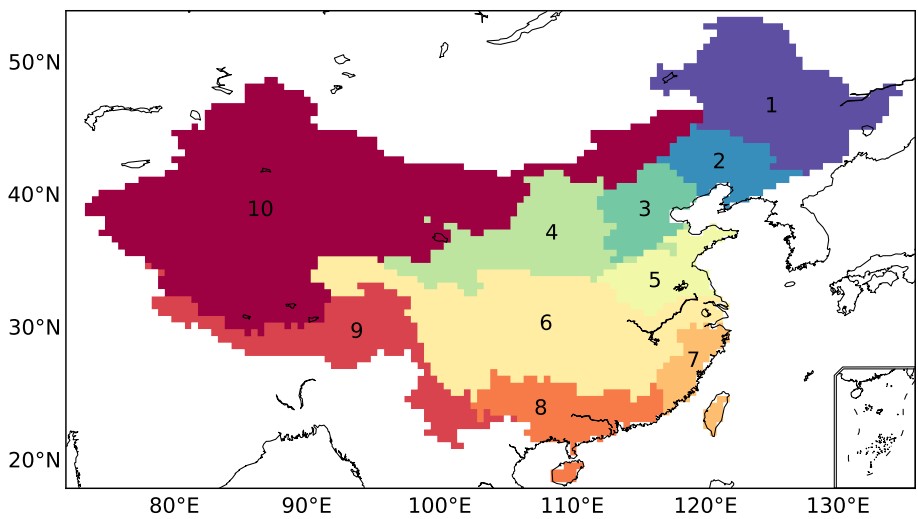

**Figure 3.** Ten subregions are identified in this study. These subregions are mainly divided as the river basins (regions 1-8) and 9 as the southwestern China and 10 as the northwestern China. The 11 represents the whole mainland.

Thirteen precipitation datasets from different sources are collected for comparison (Table 1). These datasets are categorized into three groups according to the methodologies used to generate the products, i.e., gauge-based products, merged products and General Circulation Models (GCMs). The gauge-based products (i.e., GPCC, CRU, CPC and UDEL) use observed data



from global atmospheric gauges, while the density of ground observation gauges, the representatives of the gauges and the interpolation algorithms for converting the gauge observations to grids dataset vary from product to product. CMA (provided by China Meteorological Administration) dataset uses the densest gauges and probably has the best quality to capture the spatiotemporal variations of the precipitation. But CMA is excluded when estimating the ensemble means of the gauge-based

products and chosen as the reference datasets for comparison.

Among the merged precipitation products, the CMAP, GPCP and MSWEP use different sources of precipitation data (e.g., gauge observations, satellite remote sensing, atmospheric model re-analysis). These different precipitation sources are averaged using different weights. Thus the differences among the three merged products are associated with the precipitation sources and the weight of the gauge observations. ERA-Interim is a re-analysis product, while it uses near-real-time assimilation with data

from global observations (Dee et al., 2011). Thus the forecasting model is constrained by observations and forced to follow the real system to some degree. Because of the usage of observations, ERA-interim is also belonging to the merged products.

GCM precipitation is model estimation, therefore, the physical and numerical choices will affect the accuracy of model results. In addition, observations are not used to constrain the simulations. The lack of constraints on the GCMs will cause them not following the actual synoptic variability and explore other trajectories in the solution space. Kay et al. (2015) repeatedly run

the same GCM with a very small difference in the initial conditions, and there is a spread of the model outputs after a number of time steps of running (see Figure 2 in Kay et al., 2015). Therefore, the uncertainty estimated is due to the differences in the model settings and the initial conditions. There are more than 20 datasets of GCMs, while only four are randomly taken to match the number of gauge-based products and merged products. All the datasets have been interpolated to $0.5^{\circ}$ spatial resolution to unify the spatial units and the overlap time span of all the datasets is from 1979 to 2005 for the maximum

coverage of all products.

## 2.4   What is the uncertainty means among precipitation groups

## 3   Characteristics of precipitation and model quantified uncertainties with classic metrics

### 3.1   Spatial patterns of ensemble annual precipitation

The ensemble means of the long-term annual precipitation (1979-2005), obtained by averaging the multiple datasets in the

corresponding precipitation group, are mapped in Figure 4. The long-term annual mean precipitation obtained from the CMA data is 589.8 $\mathrm{mm\ yr^{-1}}$ (1.6 $\mathrm{mm\ day^{-1}}$) over mainland China. The gauge-based precipitation has the least bias (-4.1mm $\mathrm{yr^{-1}}$, -0.7% in proportion) compared to the CMA precipitation. Precipitation in the merged products and GCMs is larger than CMA by 63.1 and 232.0 $\mathrm{mm\ yr^{-1}}$ (with the bias as +20.4% and +41.3%), respectively.

The spatial pattern of the annual precipitation shows a decreasing gradient from the southeastern China (>1600 $\mathrm{mm\ yr^{-1}}$)

to the northwestern China (<400 $\mathrm{mm\ yr^{-1}}$). All the ensemble means of the three precipitation groups capture the spatial gradient, while they have different ability to express in some details. For instance, there are some isolated areas with larger or smaller area in the CMA precipitation which could be caused by the topography (e.g., the northern Tienshan Mountain, the Qilian Mountains), while they are not shown in the gauge-based products. As we know, the precipitation gauges are mainly





**Table 1.** The precipitation datasets used in this study. Three different precipitation groups are identified according to the way the precipitation dataset is generated.

| No | Type | Name | Long name | Institute | Reference |
|---|---|---|---|---|---|
| 1 | | CMA | China Meteorological Administration dataset | China Meteorological Administration | Zhao et al. (2018) |
| 2 | | GPCC | Global Precipitation Climatology Centre | the World Climate Research Programme (WCRP) and to the Global Climate Observing System (GCOS) | Schneider et al. (2018) |
| 3 | Gauge-based | CRU TS | Climatic Research Unit Time-Series | Climatic Research Unit (CRU) / Ian Harris, Phil Jones | University of East Anglia Climatic Research Unit et al. (2017) |
| 4 | | CPC | CPC Global Unified Gauge-Based Analysis of Daily Precipitation | NCEP/Climate Prediction Center | Xie et al. (2007) |
| 5 | | UDEL | University of Delaware Air Temperature & Precipitation Global (land) precipitation and temperature | University of Delaware | Willmott and Matsuura (2012) |
| 6 | | CMAP | CPC Merged Analysis of Precipitation | NOAA CPC | Xie et al. (2003) |
| 7 | Merged Products | GPCP | Global Precipitation Climatology Project | GSFC (NASA) | Adler et al. (2018) |
| 8 | | MSWEP | Multi-Source Weighted-Ensemble Precipitation | Princeton University, Princeton, NJ, USA | Beck et al. (2017) |
| 9 | | ERA-I | ERA-Interim | European Centre for Medium-Range Weather Forecasts | Dee et al. (2011) |
| 10 | GCMs | HadCM3 | Hedley Centre Coupled Model Version 3 | Met Office Hadley Centre, UK | |
| 11 | | IPSL-CM5A-LR | | Insitut Pierre Simon Laplace, Paris, France | |
| 12 | | CMCC-CM | | Cetro Euro-Mediterraneo per I Cambiamenti | |
| 13 | | MIROC5 | | AORI, Chiba, Japan, NIES, Ibaraki, Japan, JAMSTEC, Kanagawa, Japan | |





distributed on the lower altitude and therefore, they have difficulty in captures the precipitation events over mountains. The precipitation in the merged products and the GCMs is higher than CMA in Himalayas and especially the GCMs show higher precipitation in the northern Tibet Plateau as well as the southern part of the Hengduan Mountains. These differences show the general characteristics and their difference of all the three types of precipitation products.

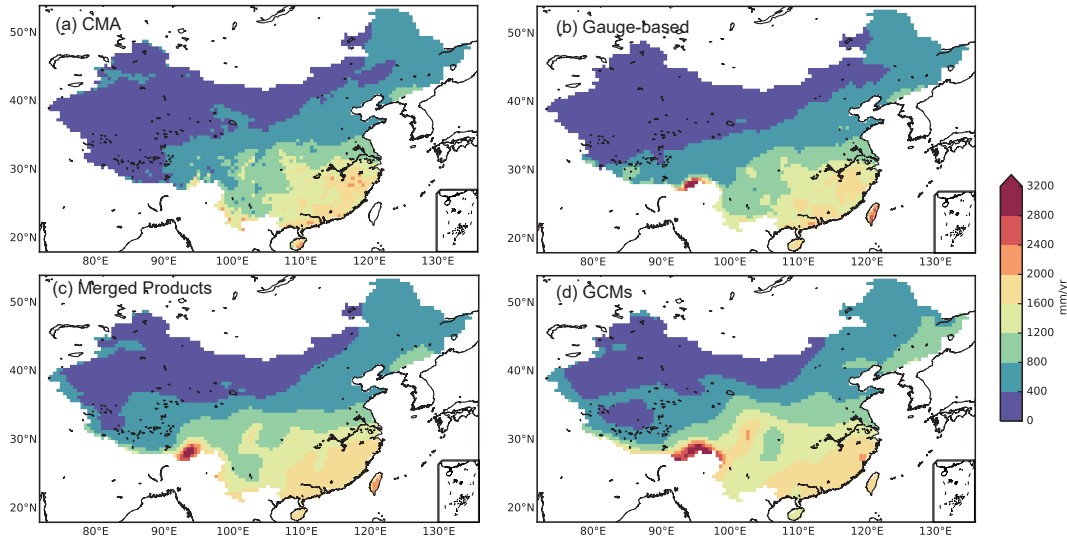

**Figure 4.** Long-term (1979-2005) annual precipitation in different precipitation groups. (a) Annual precipitation of CMA dataset, (b) ensemble means of the annual precipitation in gauge-based products excluding CMA, (c) ensemble mean of the annual precipitation of all merged products, (d) ensemble means of the annual precipitation of all GCMs. The observations in Taiwan are not included in the CMA dataset.

## 3.2 Spatial distribution of model uncertainties

In addition to differences of the long-term annual precipitation, differences are found among datasets within the same precipitation group. The spatial distribution of the model uncertainty, which is expressed as the ensemble deviation across multiple products of the annual precipitation, is calculated for each group and mapped in Figure 5.

Among the datasets based on gauge observations (Figure 5-a), the ensemble deviation value is small in most land area of China (<50 mm yr$^{-1}$). It is higher in the south of China (50-100 mm yr$^{-1}$) but the area is not continuous in space. The highest deviation occurs along the Himalayas, indicating a high variation among datasets. Regarding the merged precipitation products, the deviation shows high values (>200 mm yr$^{-1}$, Figure 5-c) in the southwestern China (e.g., the Tibet Plateau, Yunnan Province, Guangxi Province). Moderate deviation is found in the northeastern China, northern China and southeastern China. Compared to the gauge-based and merged products, the deviation among GCMs has the highest value (>400 mm yr$^{-1}$, Figure 5-e) in the southern China, indicating a significant model uncertainty of the annual precipitation between different GCMs.





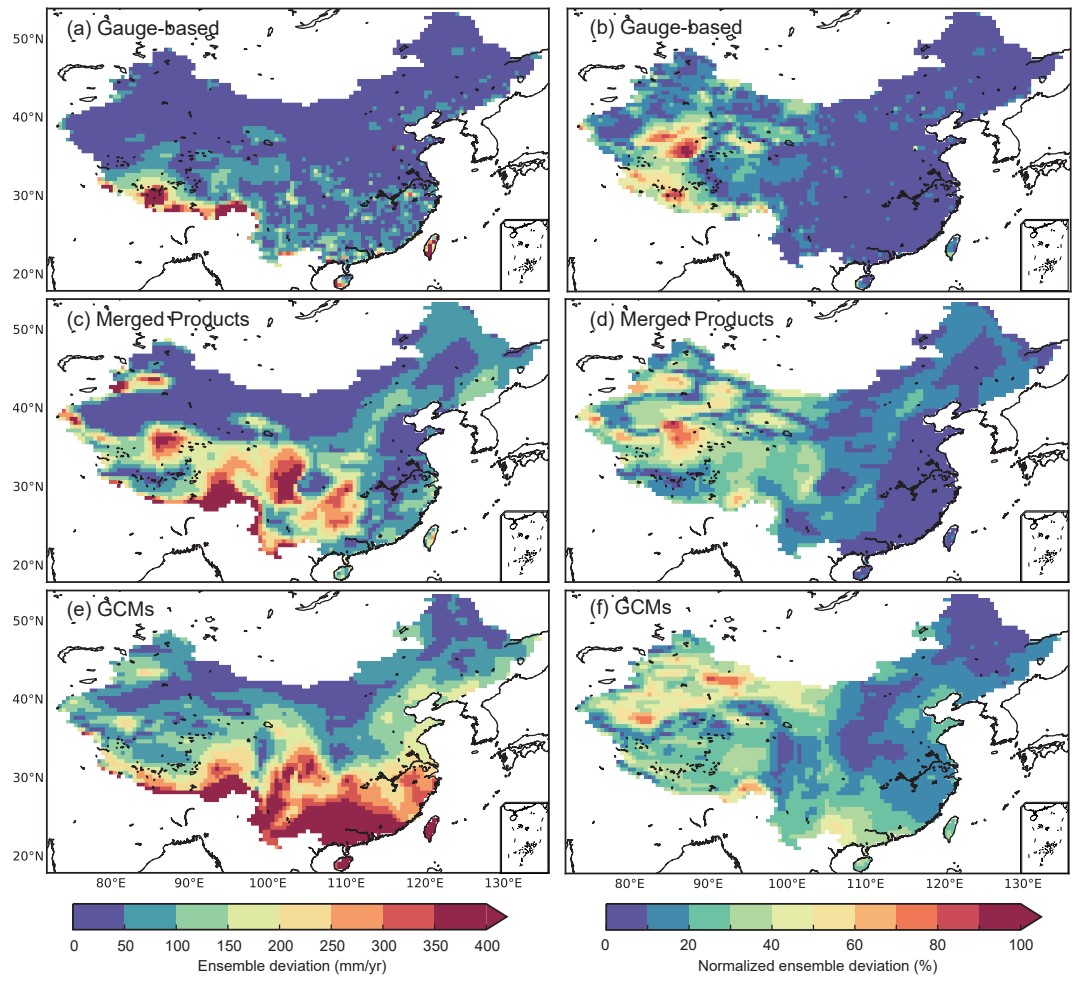

**Figure 5.** The spatial distribution of the model uncertainty, which is expressed as the ensemble deviation across multiple products of the long-term annual precipitation in each precipitation group (left panels) and the normalized value as the ratio of the ensemble deviation to the ensemble means of the datasets in corresponding group (right panels).



The ratio of the ensemble deviation to the mean value, which shows the model uncertainty with non unit, is very low (<10%, Figure 5-b) in the eastern China. While, it is higher in the western China especially in the Himalayas and the northern Tibet Plateau. Similar to that of the gauge-based products, the uncertainty in the merged products has the higher values in the west than that in the east of China (Figure 5-d). The area with the ratio less than 10% is mainly distributed in the southeastern China

and is apparently smaller than that of the gauge-based products, showing a decreasing similarity among different merged products. The area with a moderate ratio (10%-40%) increases compared to that of the gauge-based products, and the area is mostly in the middle and western China. The uncertainty estimated in the GCMs shows similar patterns in western China to that of the merged products but with higher magnitudes in the eastern China (Figure 5-f). Only the area in the northeastern and part of the middle China features small uncertainty less than 10%, and the ratio rises significantly in the southern China (e.g.,

Pearl River basin), which corresponds to the high standard deviation of the GCMs shown in Figure 5-e.

The magnitude of the ensemble deviation demonstrates the model uncertainty among different precipitation products in the same precipitation group and it shows the ability of the precipitation estimation with different methodologies. For all products, the ensemble deviation are relatively larger where the precipitation is higher, especially along the mountains and the subtropical regions.The ratio of ensemble deviation to the means showing the uncertainty more clearly is higher in the

northwestern China where the precipitation is among the lowest in China. Particularly for the gauge-based products, the higher ratio occurs where the gauge density is low and the orographic effect is apparent (e.g., the Tibet Plateau and the mountainous area). For the merged products and the GCMs, the ratio increases especially in the southeastern China, showing decreasing similarities among different GCMs. Because the ratio has taken into account both the variation and the means (which may has a systematic bias), the ratio is better than the absolute ensemble deviation to represent the uncertainty.

**3.3 Temporal evolution of model uncertainties**

Figure 5 shows the spatial distribution of the ensemble deviation among different products of the annual precipitation. However, the temporal evolution of the deviation among the various products is not captured because the temporal variation has been averaged before estimating the ensemble deviation in Figure 5. In this section, we examine the temporal evolution of model uncertainty of the regional annual precipitation across different products. The analysis is based on the ten subregions defined

in Figure 3 and the whole Chinese mainland.

The annual precipitation of each precipitation group has been normalized as the ratio to the long-term annual means of CMA (black line in Figure 6). The magnitude of the annual precipitation in the gauge-based products (blue) is similar to that of CMA except in the southwestern China (Figure 6-i) for the overestimation along the Himalayas (Figure 4). The precipitation in the merged products (green) is higher in the southwestern and northwestern China, in accordance with Figure 4-c. The annual

precipitation of the GCMs (red) is apparently higher than that of the gauge-based products or merged products for almost all regions, which agrees with the spatial patterns in Figure 4-d.

The ensemble deviation (shaded area) shown in Figure 6 represents the variations of the products in the same precipitation group at each time step. The normalized deviation facilitates the comparisons between different regions by scaling it to the means of corresponding group to obtain the width of the uncertainty range in the same scale of the y-axis. High deviations





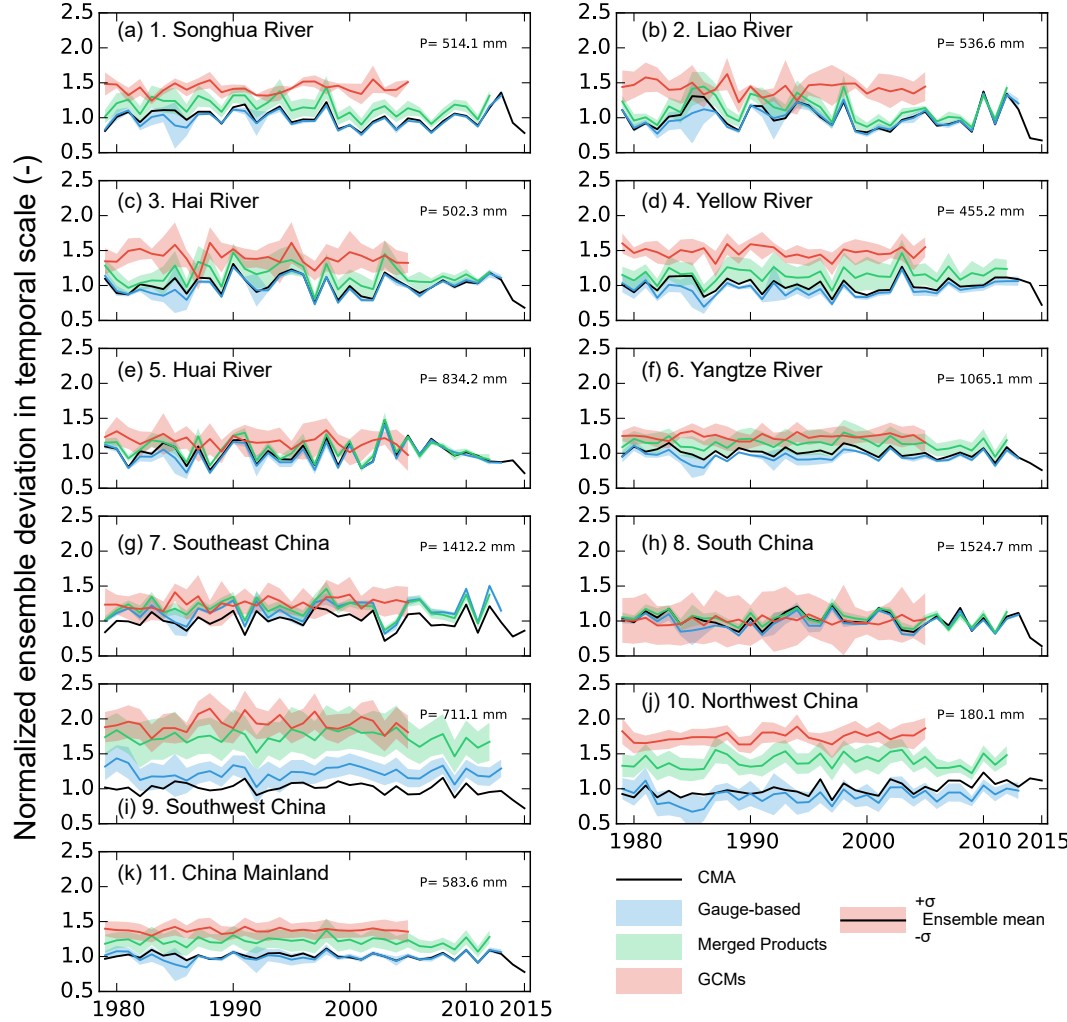

**Figure 6.** The temporal evolution of the model uncertainty, which is expressed as the normalized ensemble deviation of annual precipitation across datasets in each precipitation group for specific subregions. The value on the top right of each panel is the annual regional precipitation estimated in CMA dataset (1979-2015). The annual precipitation is normalized as the ratio to the CMA long-term annual precipitation. The shaded area represents the standard deviation of the annual precipitation in each year among the datasets within that group (divided by the annual precipitation of the corresponding group).



are found in all three precipitation groups in the southwestern China (Figure 6-i) because of the large differences along the Himalayas. The deviations among the gauge-based products and the merged products in other regions are small and getting smaller with time. It is mainly because more observations are integrated and technologies improve with time to control the data quality. A large deviation is found in the merged products in 10-northwest China (Figure 6-j) and the 4-Yellow River Basin

(Figure 6-d), where the annual precipitation is among the lowest and dry climate dominates. The model deviation of GCMs varies among regions as it is smallest in the 1-Songhua River Basin (Figure 6-a) and the 6-Yangtze River Basin (Figure 6-f), while it is among the highest in the 8-south China and the west China (9,10), agreeing with the deviation maps in Figure 5.

The temporal evolution of the gauge-based products and merged products agree well with that of the CMA dataset, while the temporal evolution of GCMs ensemble is weaker and not well correlated with that of the CMA. The main reason is that

GCMs are not constrained in their synoptic variability and the sequence of the wet and dry years can be very different from that of the observations. So a smoother result can be obtained when we build the ensemble means from the GCMs. While this is different for the gauge-based and merged products, as they have a strong co-variance and the ensemble mean preserves this co-variance.

For the entire mainland of China (Figure 6-k), the ensemble deviation remains stable for different precipitation groups. In

contrast, the annual precipitation spans the largest spatial heterogeneity in the mainland compared to those divided subregions (Figure 4). However, the spatial variation has been collapsed when estimating the regional precipitation for temporal analysis. It is therefore interesting to see how the uncertainty estimate changes when the variations in the time dimension and in the space dimension are considered together in the precipitation datasets.

### 3.4 Variations in the time and space dimensions

The precipitation varies in time and space, however, it is averaged either in the time dimension to obtain the spatial patterns of model uncertainty (Figure 5) or in the space dimension to obtain the temporal evolution of the model uncertainty (Figure 6). But the deviations in the time and space dimensions are indeed very rarely compared. Herein, the standard deviation of the temporal and spatial variations in the precipitation datasets are compared in Figure 7 in ten subregions and the Chinese mainland for different precipitation groups.

The gauge-based products provide similar annual regional precipitation to CMA over the China mainland and ten specific regions except for the region 7-southeast China (Figure 7-g) and region 9-southwest China (7-i). It might indicate the decreased ability of remote sensing, the important data source in the merged products, to estimate the precipitation amount in storms as the storms mainly contribute to the total precipitation for the two subregions. The regional precipitation is larger in merged products than that of observations and the magnitude of the deviation in GCMs is even larger except in the region 8-south

China (Figure 7-h). These results indicate the reduced ability of merged products and GCMs in reproducing the total value of the annual precipitation.

Regarding the variations in time and space dimensions, the regions 9, 10 and 11 have the largest ratio of the spatial standard deviation (to the mean), indicating the most significant spatial heterogeneity over the regions. The 7-southeast and the 3-Hai River have the smallest variations either because of the small area or because of the homogeneity in the subregion as the





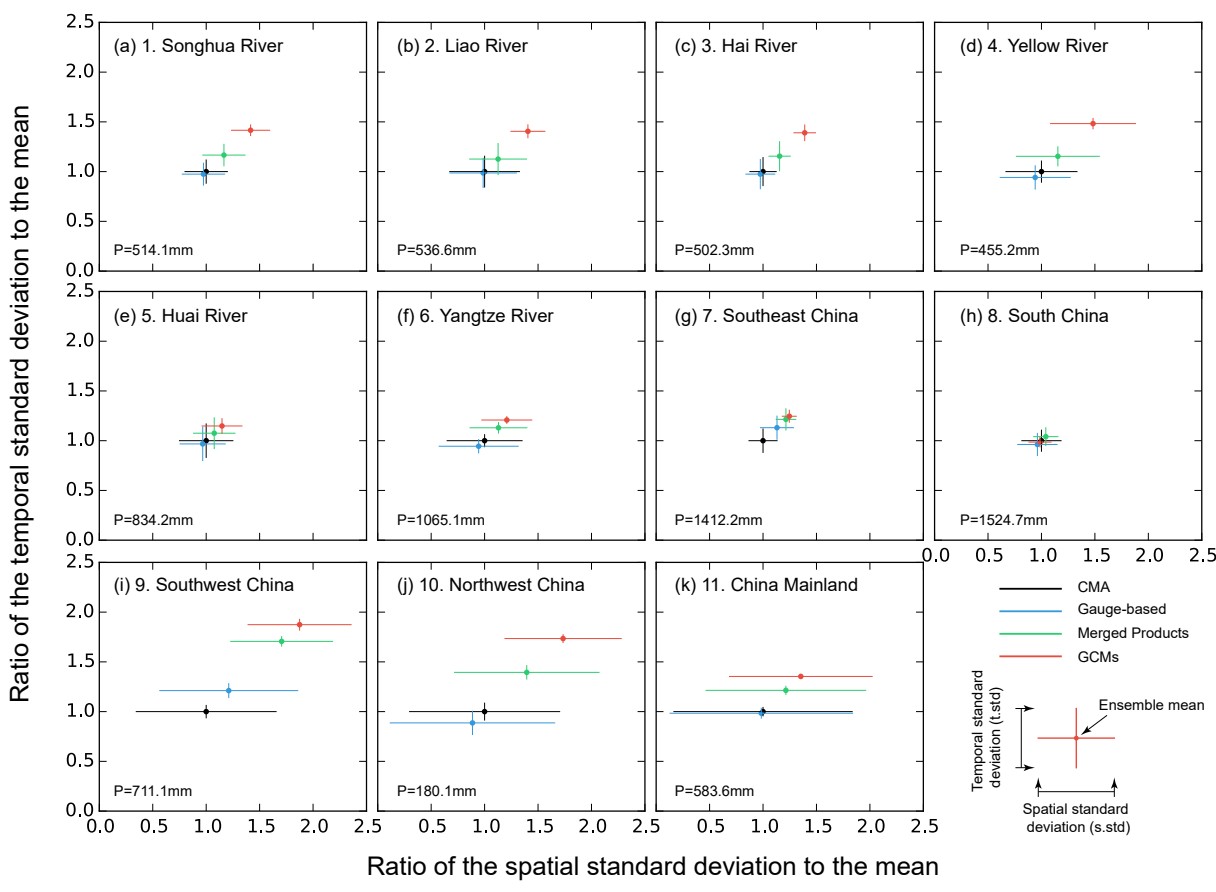

**Figure 7.** The spatial standard deviation (horizontal) and temporal standard deviation (vertical) of the annual precipitation in different precipitation groups for ten regions and the mainland China. The cross centre represents the long-term means of the regional annual precipitation. The horizontal error bar represents the spatial standard deviation (spatial variation of the long-term annual precipitation at all the grids). The vertical error bar represents the temporal standard deviation (temporal variations of region-averaged annual precipitation in different years). The P values in the left bottom is the annual precipitation of CMA.



spatial correlation is high in the area. The relative ratio of the temporal standard deviation to the spatial standard deviation is among the smallest in the regions 9, 10 and 11 ($k$=0.1, 0.12 and 0.05, respectively. $k$ is the ratio of the temporal deviation to the spatial deviation), showing an apparent difference between the variation in the time and space dimensions. While, the difference between variation in two dimensions is small in the 3-Hai River basin ($k$=1.15) and 7-southeast China ($k$=0.90), mainly because the relatively strong variability of the annual precipitation in different years.

In addition to the differences across regions, the variations in different precipitation groups are also different. Excluding the CMA dataset which only consists of one single product, the variations in the gauge-based products are higher than that of the other two groups. The difference demonstrates that on one hand the gauge-based may have the largest variation over space or on the other hand the correlation among different gauge-based products are high so that the variation is preserved when doing the ensemble. On the contrary, the GCMs have the smallest variations, either because the precipitation estimated in GCMs are more homogenous over space, or because the spatial patterns in different GCMs are not consistent and the spatial correlation is lower since there is no constrain in the GCM simulation.

## 4    Variances in precipitation products

### 4.1    Variances in three dimensions

We have introduced the general spatial and temporal characteristics of the precipitation in different groups and their variations in different dimensions in the above section. In this section, we will present the results that estimated by the newly proposed variance approach. As introduced in the methodology section, the input annual precipitation to the approach is re-organized into three dimensions as (1) **time**, 27 years from 1979 to 2005, (2) **space**, the number of 0.5° grids in a specific region and (3) **ensemble**, the number of the models in a same precipitation group (four models in all three groups).

The grand variance and the variances in three different dimensions (i.e., time, space and ensemble) for all the subregions are mapped in Figure 8. The grand variance (total value of the variance for all three dimensions) is similar for data groups of gauge-based products and the merged products (Figure 8-a,b,c), while the grand variance in GCMs is large and is approximating twice the values of the other two groups in regions 9-south China and 10-southwest China. The differences are mainly constituted by the space variance and ensemble variance (Figure 8-i,l).

The time variance ($V_t$) is the smallest among all three variance proportions, and there are very little differences of $V_t$ in the northern China (Figure 8-d,e,f). $V_t$ in the gauge-based products is higher than that in the merged products and GCMs in regions 8-southeast China and 9-south China, indicating a relatively strong temporal variation in the annual precipitation series which consists with the larger uncertainty ranges shown in Figure 6-h,i. Similar patterns of the space variance ($V_s$) are found in the gauge-based products and merged products (Figure 8-g,h), and the 7-Yangtze River basin and 9-southwest China have the largest $V_s$ because the precipitation significantly varies in space in these two regions. $V_s$ is higher in the precipitation of GCMs especially in the 9-southwest China, indicating the strong spatial heterogeneity in the GCM models over the Himalayas (Figure 8-i). The ensemble variance ($V_e$) is relatively small in most regions in gauge-based products (Figure 8-j), with the highest $V_e$ occurring in 9-southwest China. It indicates that the model variation between datasets in the observation group is





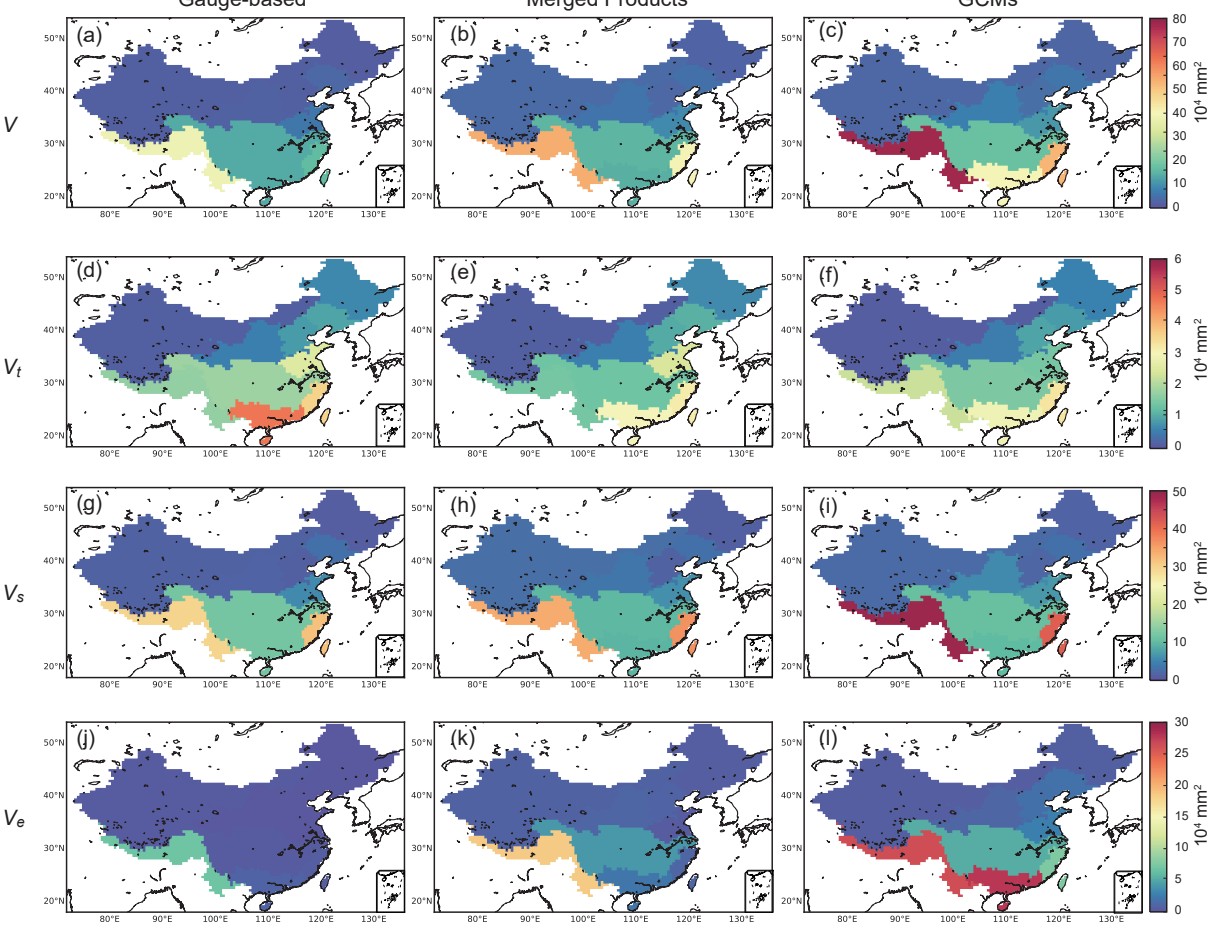

**Figure 8.** The maps of the grand variance (V) and variances in different dimensions ($V_t$, $V_s$, $V_e$) for three different precipitation groups.

small. Similar small values of $V_e$ are found in the northern regions in merged products as well as in the GCMs for the regions in the northern China, while the intra-ensemble variations are large in the south especially the 9-southwest China and 8-south China in the GCMs (Figure 8-k,l).

In conclusion, the grand variance and individual variance for each of the three different dimensions are generally larger in the dataset group consisting of GCMs. The variations for the gauge-based products and merged products are similar in values and spatial distribution. However, in addition to the variances, the uncertainty defined as the ratio of the square root of the variance to the mean (i.e., $U$, $U_t$, $U_s$, $U_e$) contains extra information of the regional means, and will be discussed in the next section.





## 4.2 Deviations in three dimensions

In contrast to the spatial patterns of the variances magnitude distributed in the ten subregions (Figure 8), the larger values of the deviation ($U = \sqrt{V}/\mu$) occur in the northwest, and lower values occur in the southern China in general (Figure 9). A possible reason is the decreasing gradient of precipitation magnitude from the southeast to the northwest (Figure 4). Although the variances are among the lowest in the northwest China, the total deviation is the highest in this region ($U$=0.89, Figure 9-a,b,c) for all three precipitation groups because of the low precipitation rate in the northwest. $U$ is relatively small in the 1-Songhua River ($U$=0.27) in the northeast and 8-South China ($U$=0.29) for the gauge-based products and 6-Yangtze River has relatively lower $U$ in the merged products and GCMs in the east part of China.

The variations in time and space dimension are natural, and they show the temporal evolution and spatial heterogeneity of the characteristics in different precipitation products (Sun et al., 2010, 2012). It is found that the $U_t$ is small and contributes very little to the total $U$, indicating the weak fluctuation of annual precipitation compared to spatial variations (Figure 9-d,e,f). The $U_t$ values are the smallest for the GCMs, in accordance with the weak temporal variations in Figure 6. The relative variance in space dimension ($U_s$) contributes the most to the total variance, especially in the northwestern China ($U_s$=0.77 for the gauge-based products, Figure 9-g). The high values indicate the strong spatial heterogeneity of precipitation in the region compared to the mean values. However, because the spatial variations characterized by GCMs in the northwestern China is less significant than other two groups, the $U_s$ for region 10-southwest China (=0.51) is smaller than that of the gauge-based and merged products.

The variations in time and space dimensions show the natural precipitation patterns but the deviation of the values at same spatiotemporal points show the ability of the products to consistently represent the spatiotemporal patterns. The relative variance in the ensemble dimension ($U_e$) shows the variations among different products in the same group. For the gauge-based products, the $U_e$ is smaller than 0.1 for regions in the eastern China, indicating that the model differences are relatively small compared to the annual means. The $U_e$ value is higher for the 9-southwest (=0.30) and 10-northwest China (=0.37), showing large variations even in the gauge-based products. For the merged products, $U_e$ is similar to that of the gauge-based products in the western China (=0.36), while it is larger in the east especially for the 6-Yangtze River and 4-Yellow River (more than two times larger than $U_e$ of the gauge-based products).

For the GCM precipitation, the uncertainty increases compared to the other two groups in the eastern regions, corresponding to the higher ensemble variations in GCM over the eastern regions shown in Figure 5. While, it decreases in 10-northwest China ($U_e$=0.25) and a possible reason is that the spatial homogeneity of the variations in the region 10-northwest China (Figure 5-f) is stronger than that of the other groups (Figure 5-b,d). In the GCMs, the highest $U_e$ occurs in the southwestern China where both the means and the variations are higher (Figure 4 and 5). As conclusion, the $U_e$ is linked with the magnitude of the model uncertainties in Figure 5 and Figure 6. It indicates that the $U_e$ is to some degree correlated to the classic metrics as the higher $U_e$ covers the grid cells or regions with higher model uncertainty.





**Figure 9.** The maps of deviations ($U$, $U_t$, $U_s$, $U_e$) estimated as the ratio of the square root of the corresponding variances (i.e., $V$, $V_t$, $V_s$, $V_e$) to the regional mean ($\mu$) for three different precipitation groups. Among the deviations, the $U_e$ is considered as the model uncertainty.





## 5 Uncertainty and metrics comparison

### 5.1 Deviation from the classic uncertainty metrics

The new estimation technique of the uncertainty $U_e$ provide a comprehensive evaluation of the uncertainty over space and time, and the values are affected by both the values and the spatial homogeneity (spatial correlations) among the different examined products. In this section, we will compare the uncertainty ($U_e$) estimated by the three-dimensional partitioning approach with the two classic metrics (defined as $N.s.std$ in Eq. 28 and $N.t.std$ in Eq.29), to explain how these three metrics are related and differ with each other.

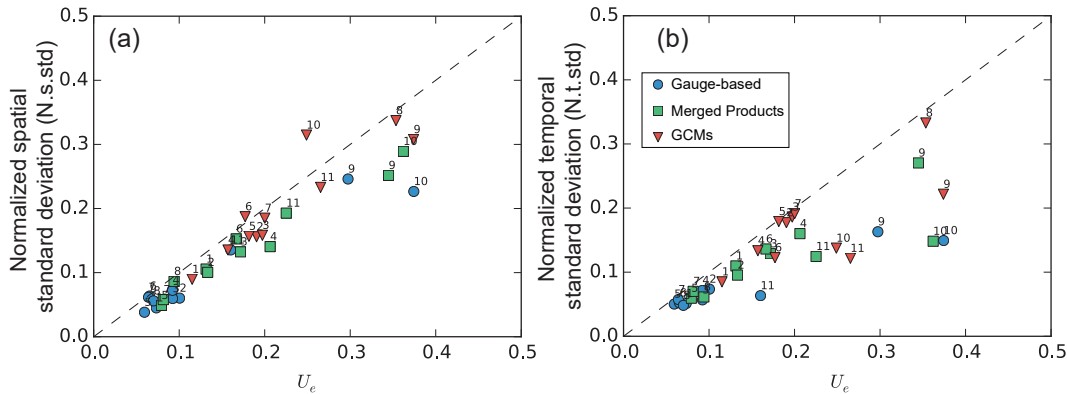

**Figure 10.** The relation of the $U_e$ to two classic metrics as (a) the normalized spatial standard deviation - $N.s.std$ and (b) the normalized temporal standard deviation - $N.t.std$.

As shown in Figure 10, $U_e$ is correlated to both the $N.s.std$ and $N.t.std$, especially when $U_e$ is smaller than 0.2 where the regions from 1 to 8 are generally included for all three precipitation groups. The $U_e$ is in general larger than the $N.s.std$ and $N.t.std$ for the products. And the deviation is because the variations of the other dimension have collapsed when calculating the spatial deviation (or temporal deviation). For the regions 9, 10 and 11, the values of the $N.s.std$ and $N.t.std$ deviate the most from the 1:1 line of the $U_e$. Taking subregion 9-southwest China in the gauge-based products as an example, the temporal variance is 62.4 mm yr$^{-1}$ while the spatial variance is 571.8 mm yr$^{-1}$ (Figure 7-i). The difference between $N.s.std$ and $U_e$ is 0.058 (=0.297-0.239, changing ratio is 24.3%) when the temporal variation is collapsed while the difference between $N.t.std$ and $U_e$ is 0.126 (=0.297-0.171, changing ratio is 73.4%) when the spatial variation, which is significantly larger than the temporal variation, is collapsed.

These regions (9, 10, 11) feature strong spatial heterogeneities (Figure 7-i,j,k) in the annual mean precipitation (Figure 4). The spatial correlation of the annual precipitation and the temporal correlation of the regional precipitation is also weaker in these three regions than other regions (*not shown in the results*). The averaging will cause a significant smooth of the datasets when the spatial correlation among datasets are very low. The spatial variation across space is also significantly higher than temporal variations (Figure 7). Because the estimation of $N.t.std$ needs the averaging in spatial dimension which may include



more information than that in the time dimension, the deviation between $N.t.std$ and $U_e$ (Figure 10-b) is larger than that between $N.s.std$ and $U_e$ (Figure 10-a). The priority of the precipitation types also change from the model dominated (the model uncertainty in GCMs are larger than the other) to the region dominated (uncertainty in specific regions 9,10,11 are larger than other regions no matter in which precipitation data). This indicates that difference of model uncertainty over space has been reflected in the new uncertainty estimation $U_e$.

## 5.2 Decomposition of the ensemble uncertainty

We now decompose the ensemble variance to explore the possible reason for the deviation of $U_e$ from the $N.s.std$ and $N.t.std$. As shown in Eq. (26), the ensemble variance is formulated as

$$V_e = \frac{1}{3}[\frac{\overline{\sigma_{e\_t}^2} + \overline{\sigma_{e\_s}^2}}{2} + \overline{\sigma_e^2} + \sigma_e^2(\mu_{ts})] \tag{30}$$

It combines four elements which contribute to the variation of different values across the ensemble dimension (i.e., the variance of original temporal and spatial values - $\overline{\sigma_e^2}$, of the temporal mean - $\overline{\sigma_{e\_t}^2}$, of the spatial mean - $\overline{\sigma_{e\_s}^2}$ and of the grand mean - $\sigma_e^2(\mu_{ts})$). Among which, the $\overline{\sigma_{e\_t}^2}$ is the mean of the square of spatial standard deviation in Figure 5-a,c,e for all grids in a specific region and $\overline{\sigma_{e\_s}^2}$ is the mean of the square of the temporal standard deviation in Figure 6 for each time step in a specific region. These two components are related to the two classic metrics $N.s.std$ (Eq. 28) and $N.t.std$ (Eq. 29), respectively.

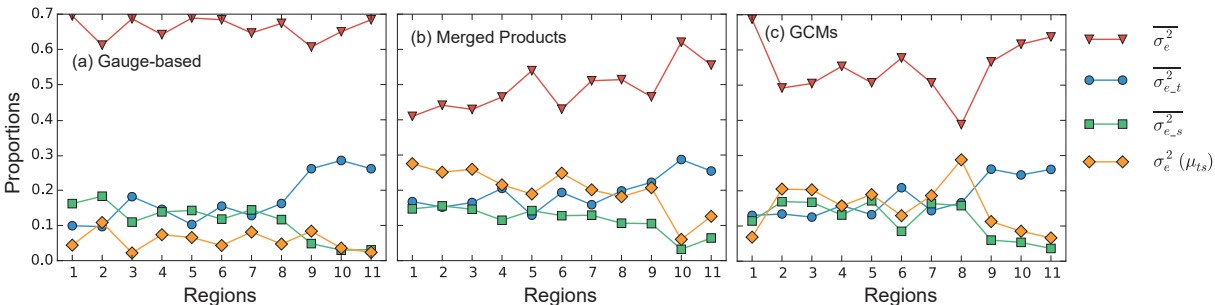

**Figure 11.** The proportion of the four components in Eq. (30) to the $V_e$ in three precipitation groups, (a) gauge-based products, (b) merged products and (c) GCMs. The contribution are normalized so that the sum of them is 1.0 for each region. Among the four components, the $\overline{\sigma_{e\_t}^2}$ and $\overline{\sigma_{e\_s}^2}$ are associated with the two classic metric $N.s.std$ and $N.t.std$, respectively.

By decomposing the Eq. (30), the contributions of the four components to the ensemble variance ($V_e$) are shown in Figure 11. For all three precipitation groups, $\overline{\sigma_e^2}$ is the dominant component simply because all the information on variations among the original datasets is retained in the uncertainty estimation. While, the other three components are estimations after averaging is performed in time, space or the full spatiotemporal dimensions, which indicates a loss of information. The contribution of the $\overline{\sigma_{e\_t}^2}$ and $\overline{\sigma_{e\_s}^2}$ is approximating 0.15 for regions from 1 to 8. While the $\overline{\sigma_{e\_t}^2}$ increases for the region 9, 10 and 11, indicating that the spatial heterogeneity is significant for these regions. On the contrary, $\overline{\sigma_{e\_s}^2}$ decreases because the spatial averaging has





collapsed the spatial variations. The very small contribution of $\overline{\sigma^2_{e\_s}}$ related to $N.t.std$ is the cause for larger deviations between $N.t.std$ and $U_e$ (Figure 10-b).

Although all the components can be used as metrics for evaluating the variations among multiple datasets, there are limitations for each of the variations. For the variation of temporal mean $\overline{\sigma^2_{e\_t}}$ and spatial mean $\overline{\sigma^2_{e\_s}}$, the collapse of a dimension has

ignored part of the information (also introduced in the Introduction). Moreover, the variation of the grand mean $\sigma^2_e(\mu_{ts})$ has ignored both the temporal variability and spatial heterogeneity, which further decreases its applicability in uncertainty assessment. The variation $\overline{\sigma^2_e}$ is estimated based on the original data without averaging, and thus represents the most information. However, it cannot account for the systematic uncertainty (bias in the mean values) which is expressed as $\sigma^2_e(\mu_{ts})$.

Therefore, all the four elements represent the model variations from different aspects and neither of the single element

is able to represent all the others. Integration of different components ($V_e$) is therefore a solution to indicate all metrics to different degrees. What is interesting is that the variability of the proportions of $\overline{\sigma^2_{e\_t}}$ and $\overline{\sigma^2_{e\_s}}$ (or $\overline{\sigma^2_e}$ and $\sigma^2_e(\mu_{ts})$) are opposite and the sum of their proportions is stable around 0.3 (or 0.7). This indicates a complementary relation between the two pairs of elements ($\overline{\sigma^2_{e\_t}}$ & $\overline{\sigma^2_{e\_s}}$; $\overline{\sigma^2_e}$ & $\sigma^2_e(\mu_{ts})$). On the other word, some of the information is ignored in one of the element but remained in the other one within the same pair. And therefore it indicates that the variation in the time dimension and that in

the space dimension should be considered together as done in the estimation of the ensemble variance ($V_e$). The normalized metric ($U_e$) derived from the integrated variation ($V_e$), which has better ability to demonstrate the uncertainties compared to the classic metrics, should be a better choice for the uncertainty analysis.

### 5.3  Metrics differences in value and proportion

Figure 10 shows that the $U_e$ is generally higher than the uncertainty identified by the two classic metrics, $N.s.std$ and $N.t.std$.

Figure 12 then summaries the magnitude of the changes from the classic metrics to the new uncertainty estimation identified by $U_e$. We can find that the two classic metrics generally underestimate the uncertainty by around 0.03 (Figure 12-a). The variation of the underestimation of $N.t.std$ is larger than that of the $N.s.std$, showing a larger deviation between the $U_e$ with $N.t.std$. Applying the new uncertainty metric will increase the estimation of uncertainty by around 20-40% for half of the cases compared to the $N.s.std$ (Figure 12-b). For nearly 25% of the cases, the new $U_e$ increases the estimation of uncertainty

by more than 50%. In the extreme cases, $U_e$ is larger than twice $N.t.std$ (Figure 12-b). The results show that the known uncertainty estimated by the two classic metrics, which have been widely applied to climatic analysis, have underestimated the uncertainty among different models / datasets, which has been assumed when introducing the peculiarity of the new method. The underestimation may especially occur for assessment of temporal evolution of the uncertainties ($N.t.std$), which is very commonly seen in scientific reports and articles to illustrate the temporal evolution of the variables of interest.



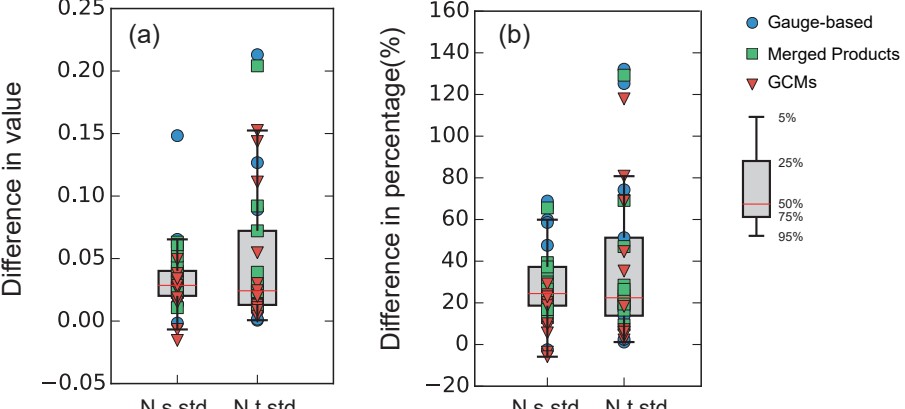

**Figure 12.** The changes in (a) value and (b) percentage when using $U_e$ as the new uncertainty metric compared to classic metrics $N.s.std$ (Eq. 28) and $N.t.std$ (Eq. 29).

## 6 Discussion and Conclusion

### 6.1 Features and applicability of the approach

The proposed variance partitioning approach works in three dimensions, and it is able to use all of the information in the time and the space dimensions among the multiple ensemble members. The proposed $U_e$ estimation technique is especially suitable

for the overall assessment of the variations among multiple datasets over a certain period and over a specific space. Though, the compensation is that the $U_e$ technique cannot provide the temporal evolution or spatial heterogeneity for users' consideration. However, in most cases we would like to know the general performance of the ensemble models with single estimate. The two classic metrics (eqs.28 and 29) are also single values but their estimation has averaged the variations which means a loss of informations.

The results of the partitioning approach can be affected by the choice of the time step intervals. For example, the time variation or time variance proportion will significantly increase if the time interval is chosen as one month. The inter-annual variation of precipitation will result in higher $V_t$ and lower $V_s$ or $V_e$. It depends how significant the inter-annual variability is compared to the intra-annual variations.

The proposed approach has a flexible structure that potentially deals with different problems from global to regional di-

mensions. The time dimension can consider intervals from daily, monthly, annual or to decadal analysis in different scopes. The ensemble dimension is applicable from 2 members (i.e., model evaluation between simulations and observations) to any number of multi-models (consensus evaluation, Tebaldi et al., 2011; McSweeney and Jones, 2013). The present approach is applicable to any variables that are organized in the three dimensions such as climatic variables (e.g., temperature, evaporation), hydrological variables (e.g., soil moisture, runoff) or environmental variables (e.g., drought index). Based on these advantages,

the three-dimensional partitioning approach can widely be applied in the hydro-climatic analysis.





## 6.2  Conclusion

A new three-dimensional partitioning approach is proposed in this study to assess the model uncertainties among multiple datasets. The new uncertainty metric ($U_e$) is estimated with an overall consideration of temporal and spatial variations among the ensemble products. Results show that $U_e$ is generally larger than the classical uncertainty metrics $N.s.std$ and $N.t.std$ which require a collapse in either of the time or space dimension. The deviation occurs where the spatial variations are significant but being averaged in $N.t.std$ estimation. The decomposing of the $V_e$ shows the complementary relation of the two classic metrics and therefore the new uncertainty estimation ($U_e$, derived from $V_e$) technique is a more comprehensive estimation way of uncertainty for multiple datasets.

Thirteen precipitation datasets generated by different methodologies are categorized into three groups (i.e., gauge-based products, merged products and GCMs) and the model uncertainty in the ensemble products in the same group is analyzed with the new and two classic uncertainty metrics. The GCMs are identified with the largest model uncertainty with the classical metrics in most regions, while the new estimation $U_e$ indicates the largest model uncertainty occurs in specific regions no matter in which precipitation group. The spatial heterogeneity of the model uncertainty over space has been represented well in the new uncertainty metric. Thus, the overall model uncertainty ($U_e$) is a new uncertainty estimate which involves more information and should receive more attention in the uncertainty assessment field.

*Data availability.*  The CMA dataset is available through website http://data.cma.cn/data/detail/dataCode/SURF_CLI_CHN_PRE_MON_ GRID_0.5/ (Zhao et al., 2018); The GPCC dataset is available at https://doi.org/10.5676/DWD_GPCC/FD_M_V2018_050 (Schneider et al., 2018) ; The CRU dataset is available at https://doi.org/10.5285/edf8febfdaad48abb2cbaf7d7e846a86 (University of East Anglia Climatic Research Unit et al., 2017); The CPC dataset is available at http://ftp.cpc.ncep.noaa.gov/precip/CPC_UNI_PRCP/ (Xie et al., 2007); The UDEL datasets is available at http://climate.udel.edu/data (Willmott and Matsuura, 2012); The CMAP dataset is available at https://www.esrl. noaa.gov/psd/data/gridded/data.cmap.html (Xie et al., 2003); The GPCP dataset is available at https://www.esrl.noaa.gov/psd/data/gridded/ data.gpcp.html (Adler et al., 2018); The MSWEP dataset is available at http://gloh2o.org (Beck et al., 2017); The ERA-I dataset is available at https://www.ecmwf.int/en/forecasts/datasets/archive-datasets/reanalysis-datasets/era-interim (Dee et al., 2011); The CMIP5 GCMs products are available at https://esgf-node.llnl.gov/projects/esgf-llnl/.

## Appendix A:  The algorithms for different expressions in the methodology

Zone A:

A1: $\mu_t[s, e; n \times l]; \mu_t[j, k] = \frac{1}{m} \sum_{i=1}^{l} z_{ijk}$

A2: $\mu_s[e, t; l \times m]; \mu_s[k, i] = \frac{1}{n} \sum_{j=1}^{l} z_{ijk}$

A3: $\mu_e[t, s; m \times n]; \mu_e[i, j] = \frac{1}{l} \sum_{k=1}^{l} z_{ijk}$

Zone B:

B1: $\sigma_t^2[s, e; n \times l]; \sigma_t^2[j, k] = \frac{1}{m} \sum_{i=1}^{l} (z_{ijk} - \mu_t[j, k])^2$





B2: $\sigma_s^2[e,t;l \times m]; \sigma_s^2[k,i] = \frac{1}{n}\sum_{j=1}^{l}(z_{ijk} - \mu_s[k,i])^2$

B3: $\sigma_e^2[t,s;m \times n]; \sigma_e^2[i,j] = \frac{1}{l}\sum_{k=1}^{l}(z_{ijk} - \mu_e[i,j])^2$

Zone C:

C1: $\sigma_{t\_s}^2[e;l]; \sigma_{t\_s}^2[k] = \sigma^2(\mu_s[k,:])$

C2: $\sigma_{t\_e}^2[s;n]; \sigma_{t\_e}^2[j] = \sigma^2(\mu_e[:,j])$

C3: $\sigma_{s\_t}^2[e;l]; \sigma_{s\_t}^2[k] = \sigma^2(\mu_t[:,k])$

C4: $\sigma_{s\_e}^2[t;m]; \sigma_{s\_e}^2[i] = \sigma^2(\mu_e[i,:])$

C5: $\sigma_{e\_t}^2[s;n]; \sigma_{e\_t}^2[j] = \sigma^2(\mu_t[j,:])$

C6: $\sigma_{e\_s}^2[t;m]; \sigma_{e\_s}^2[i] = \sigma^2(\mu_s[:,i])$

Zone D:

D1: $\mu_{et}[s;n]; \mu_{et}[j] = \frac{1}{lm}\sum_{k=1}^{l}\sum_{i=1}^{m} z_{ijk}$

D2: $\mu_{se}[t;m]; \mu_{se}[i] = \frac{1}{nl}\sum_{j=1}^{n}\sum_{k=1}^{l} z_{ijk}$

D3: $\mu_{ts}[e;k]; \mu_{ts}[k] = \frac{1}{mn}\sum_{i=1}^{m}\sum_{j=1}^{n} z_{ijk}$

Zone E:

E1: $\sigma_{et}^2[s;n]; \sigma_{et}^2[j] = \frac{1}{lm}\sum_{k=1}^{l}\sum_{i=1}^{m}(z_{ijk} - \mu_{et}[j])^2$

E2: $\sigma_{se}^2[t;m]; \sigma_{se}^2[i] = \frac{1}{nl}\sum_{j=1}^{n}\sum_{k=1}^{l}(z_{ijk} - \mu_{se}[i])^2$

E3: $\sigma_{ts}^2[e;l]; \sigma_{st}^2[k] = \frac{1}{mn}\sum_{i=1}^{m}\sum_{j=1}^{n}(z_{ijk} - \mu_{ts}[k])^2$

Zone F:

F1: $\sigma_t^2(\mu_{se}) = \frac{1}{m}\sum_{i=1}^{m}(\frac{1}{nl}\sum_{j=1}^{n}\sum_{k=1}^{l} z_{ijk} - \frac{1}{m}\sum_{i=1}^{m}(\frac{1}{nl}\sum_{j=1}^{n}\sum_{k=1}^{l} z_{ijk}))^2$

F2: $\sigma_s^2(\mu_{et}) = \frac{1}{n}\sum_{j=1}^{n}(\frac{1}{lm}\sum_{k=1}^{l}\sum_{i=1}^{m} z_{ijk} - \frac{1}{n}\sum_{j=1}^{n}(\frac{1}{lm}\sum_{k=1}^{l}\sum_{i=1}^{m} z_{ijk}))^2$

F3: $\sigma_e^2(\mu_{ts}) = \frac{1}{l}\sum_{k=1}^{l}(\frac{1}{mn}\sum_{i=1}^{m}\sum_{j=1}^{n} z_{ijk} - \frac{1}{l}\sum_{k=1}^{l}(\frac{1}{mn}\sum_{i=1}^{m}\sum_{j=1}^{n} z_{ijk}))^2$

The $t, s, e$ in the algorithms represents the three dimensions **time**, **space** and **ensemble**, with the size of $m, n, l$, respectively. Each expression is shown with its size and the meaning of each dimension. For example, for the A1: $\mu_t[s,e;n \times l]$, the $\mu_t$ has a size of $n \times l$. The first axis represents the space dimension, and the second is the ensemble dimension. While C1 ($\sigma_{t\_s}^2[e;l]$) has only one ensemble dimension with its size as $l$. F1 ($\sigma_t^2(\mu_{se})$) is only a single value.

*Author contributions.* XZ initialized the ideas presented in this paper with supervising from JP and TY. XZ prepared the simulations, the figures and the manuscript. CSH participated in the data preparation. All authors contributed to the discussion and revising the paper.

*Competing interests.* The authors declare that they have no conflict of interest.



*Acknowledgements.* This study was supported by the National Natural Science Foundation of China (grant nos. 41561134016 and 51879068); the CHINA-TREND-STREAM French national project (ANR grant no. ANR-15-CE01-00L1-0L); the National Key Research and Development Program (2018YFC0407900) and the China Scholarship Council (CSC, 201506710042). The work was supported by computing resource of the IPSL ClimServ cluster at École Polytechnique, France.



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
