# Peer review of "A new uncertainty estimation approach with multiple datasets and implementation for various precipitation products"

_Hydrology and Earth System Sciences, 2019_

## Referee Comment (RC1) · Anonymous Referee #1 · 27 Feb 2019

General comments The study proposed a new uncertainty estimation that takes into account both spatial, temporal, and model uncertainties. The authors then compared the new uncertainty values with two classic uncertainty metrics and demonstrated the comprehensiveness of the new metric. As the new uncertainty estimation still bears some similarity with the two classical metrics, it could be used as an alternative metric. The reviewer recommends minor revision.

Specific comments Section 2.4 is missing.

L16 on page 7: change "Similarity" to "Similarly".

L1 on page 9: change "can also be expressed as the normalized" to "can also be

normalized"

L13 on page 9: change "more natural" to "more proper".

L1 on page 10: the use of "global atmospheric gauges" is not proper, change to "global precipitation gauges" instead. Change "representatives" to "representativeness".

L2 on page 10: change "grids dataset" to "gridded dataset". Change "provided by" to "stands for".

L28 on page 10: the percent biases are calculated wrongly. Suppose you use CMA annual precipitation as the base, then the percent biases are: (63.1/589.8)x 100% = 10.7%, and (232/589.8)x 100% =39.3%, respectively.

L31-32 on page 10: Do you mean some areas have abrupt precipitation changes rather than following the general gradient? The use of "isolated areas" is confusing to me.

L4 on page 12: the description is confusing.

L1 on page 14: change "non unit" to "no unit".

L18 on page 14: change "which may has" to "which may have".

L8-9 on page 16: Figure 6i and 6j do not agree well for gauge-based and merged products, so it is not proper to generalize like this sentence.

L15 on page 16: change "divided" to "categorized" or something similar.

L25-28 on page 16: The comparison between gauge-based products and CMA was mentioned firstly according to Figure 7, and then the reason for the discrepancy between the merged products and CMA was discussed. The transition was missing in between.

L6-12 on page 18: Are the standard deviations of each precipitation data group related to the number of data products that you chose?

L23-33 on page 18: It may be better to denote the subregion numbers in Figure 8, so

the audience do not need to go back and forth to identify the subregions.

L31-32 on page 20: It seems that higher U_s also correlated to regions with higher model uncertainty in Figure 9 g-i.

---

## Author Comment (AC1) · 5 Apr 2019

Dear Reviewer,

Thank you so much for your nice comments and kind suggestions! We will reply simply to your questions and comments in this file. In the meantime, we will revise the manuscript and submit it once all the comments are collected.

General comments The study proposed a new uncertainty estimation that takes into account both spatial, temporal, and model uncertainties. The authors then compared the new uncertainty values with two classic uncertainty metrics and demonstrated the comprehensiveness of the new metric. As the new uncertainty estimation still bears some similarity with the two classical metrics, it could be used as an alternative metric. The reviewer recommends minor revision.

Specific comments Section 2.4 is missing.

RE: Yes, sorry for the mistake. Instead, we moved the statements for "underlying of the uncertainties" to previous section. The section 2.4 will be removed in the revised manuscript.

L16 on page 7: change "Similarity" to "Similarly".
L1 on page 9: change "can also be expressed as the normalized" to "can also be normalized"
L13 on page 9: change "more natural" to "more proper".
L1 on page 10: the use of "global atmospheric gauges" is not proper, change to "global precipitation gauges" instead. Change "representatives" to "representativeness".
L2 on page 10: change "grids dataset" to "gridded dataset". Change "provided by" to "stands for".
L28 on page 10: the percent biases are calculated wrongly. Suppose you use CMA annual precipitation as the base, then the percent biases are: (63.1/589.8)x 100% = 10.7%, and (232/589.8)x 100% =39.3%, respectively.

RE: We have corrected the above items.

L31-32 on page 10: Do you mean some areas have abrupt precipitation changes rather than following the general gradient? The use of "isolated areas" is confusing to me.

RE: Thanks for your correction. It is exactly what you're mentioning.

L4 on page 12: the description is confusing.

RE: L4: "*These differences show the general characteristics and their difference of all the three types of precipitation products.* "

The statement is based on Figure 4. Figure 4 shows the precipitation patterns of three different precipitation groups. Based on the comparisons among the precipitation datasets, the characteristics of each one have been clarified in the words before this sentence. We revised the sentence as "*These differences show the general characteristics of the three types of precipitation products.*"

L1 on page 14: change "non unit" to "no unit".
L18 on page 14: change "which may has" to "which may have".

RE: We have corrected the above items.

L8-9 on page 16: Figure 6i and 6j do not agree well for gauge-based and merged products, so it is not proper to generalize like this sentence.

RE:  L8-9: "*The temporal evolution of the gauge-based products and merged products agree well with that of the CMA dataset, while the temporal evolution of GCMs ensemble is weaker and not well correlated with that of the CMA.*"

We also found that the average value of the merged-products are higher than the CMA data in Figure 6-I and Figure 6-j (L35 on P14 & L4 on P16). However, regarding the temporal variations (which can be quantified as the correlation), both the gauge-based products and merged products show good correlation with the CMA for all the subregions including (i) southwest and (j) northwest China.

L15 on page 16: change "divided" to "categorized" or something similar.

RE: Ok.

L25-28 on page 16: The comparison between gauge-based products and CMA was mentioned firstly according to Figure 7, and then the reason for the discrepancy between the merged products and CMA was discussed. The transition was missing in between.

RE: Thanks, we will move the explanations somewhere else to increase the readability.

L6-12 on page 18: Are the standard deviations of each precipitation data group related to the number of data products that you chose?

RE: Yes, but when the number of data products increases to a certain number (4-5 according to our experiments on the GCM products) the standard deviations (or the variance proportions which come later) will become stable. However, we limit the used products for 4 because we don't have enough independent gauge-based products or merged products.

L23-33 on page 18: It may be better to denote the subregion numbers in Figure 8, so the audience do not need to go back and forth to identify the subregions.

RE: OK, thanks. All the maps of subregions will be numbered.

L31-32 on page 20: It seems that higher U_s also correlated to regions with higher model uncertainty in Figure 9 g-i.

RE: Yes. The U_s (the third row in Figure 9) has similar patterns with that of the U_e (the fourth row in Figure 9), this is because in the original datasets, the regions with higher model uncertainty are always feature higher spatial heterogeneities (shown in Figure 5). The U_s and U_e just separate and quantify the uncertainty (or heterogeneity) of the two dimensions.

As we state in L19-20 on P7, we will focus our discussions on the U_e, we emphasize very little about the U_s or U_t. We will see if in somewhere we need to add such explanations on the similarities among different variance uncertainties.

Xudong Zhou

---

## Referee Comment (RC2) · Anonymous Referee #2 · 11 Jul 2019

The manuscript entitled "A new uncertainty estimation technique for multiple datasets and its application to various precipitation products" introduced the variance partitioning method into uncertainty quantification of ensemble precipitation datasets, which considers both temporal and spatial uncertainties, and thus established a more comprehensive uncertainty metric as compared with the classic metrics. The deviation of the mathematical framework is rigorous and complete, while lots of work, including various precipitation products in multiple regions, was conducted in the validation of the new metric. On the other hand, some theoretical questions are needed to be explained clear and the English writing of this manuscript needs improvement. The detailed problems are listed as follows: (1) According to the definition of the new uncertainty

[Figure]

metric, it's one of components partitioned from the SST over time, space and ensembles. Thus, the new uncertainty Ve interacted with other components (Vt and Vs), as the authors discussed in Section 6.1. Given an ensemble of precipitation, if we replace one year's data to make the inter-annual variation larger, then the Ve obtained will correspondingly decrease. However, this decrease of Ve resulted from regular temporal variation instead of variability of ensemble precipitation datasets. In summary, how to separate the influence of normal spatio-temporal variation from the ensemble variability representing the new uncertainty estimation? In addition, is the classical N.t.std or N.s.std affected by the same temporal or spatial variation? (2) Following Comment (1), the authors should clearly define the reasonable variation resulted from temporal dynamic or spatial heterogeneity and variability associated with uncertainty investigated in the present study of biased ensemble precipitation datasets. Also, as the interaction between the spatio-temporal variation and ensemble data variability exists, is it part of the uncertainty Ve? (3) The authors said the new uncertainty metric Ve contained both temporal and spatial uncertainties at the same time, while the classical metrics (N.t.std or N.s.std) contained only one source of uncertainties. Why the comparison of Ve with classical metrics was conducted by using each of classical metrics rather than the sum of N.t.std and N.s.std? (4) Many literatures in the field of hydro-meteorology have studied on the variance decomposition method in multi-source uncertainty investigation. What's the difference between the new uncertainty estimation partitioned from grand variance and previous studies should be highlighted in Abstract and Introduction. (5) There exist many grammar mistakes in the manuscript. For example, "an new uncertainty ..." in Abstract should be "a new uncertainty ...", the expression of "which has been included the model variation" in Abstract is wrong, "of" was omitted after "because" in Line 30 on Page 2. In addition, please check Line 16 on Page 5 and Line 8 on Page 25. (6) Despite the grammar mistakes, multiple improper or incomplete English expressions also tended to hinder the readability of the paper. For example, the mean precipitation value in Line 26 on Page 10 may be not only derived from "The long-term annual mean precipitation" but also from the lumping of spatial grids? To make reviewers and readers fully understand this study, the English should be improved considerably throughout the manuscript. (7) The gauge precipitation provided by CMA was taken as the benchmark. Although the CMA data was excluded from gauge-based group, other gauge-based products also contained part of the gauge data from CMA. This is expected to clearly state. Were the gauge-based data downloaded in grid or gauge format? Are all the precipitation data in daily time scale? (8) Why is there no content in the section of 2.4? (9) In Figure 6, since the curves plotted represented the uncertainty, what does the band of ±standard deviation around the curves mean, the uncertainty of uncertainty? Please explain. (10) In Figure 12, the quantile of the box is increasing from bottom to top for normal boxplots, while there is inverse order of quantiles here. Please check it.

---

## Author Comment (AC2) · 31 Jul 2019

Dear Reviewer:

Thank you so much for your kind suggestions, we reply to your comments shortly here and we will make corresponding changes in the revised manuscript.

The manuscript entitled "A new uncertainty estimation technique for multiple datasets and its application to various precipitation products" introduced the variance partitioning method into uncertainty quantification of ensemble precipitation datasets, which considers both temporal and spatial uncertainties, and thus established a more comprehensive uncertainty metric as compared with the classic metrics. The deviation of the mathematical framework is rigorous and complete, while lots of work, including various precipitation products in multiple regions, was conducted in the validation of the new metric. On the other hand, some theoretical questions are needed to be explained clear and the English writing of this manuscript needs improvement. The detailed problems are listed as follows:

(1) According to the definition of the new uncertainty metric, it's one of components partitioned from the SST over time, space and ensembles. Thus, the new uncertainty Ve interacted with other components (Vt and Vs), as the authors discussed in Section 6.1. Given an ensemble of precipitation, if we replace one year's data to make the inter-annual variation larger, then the Ve obtained will correspondingly decrease.

Reply: Yes, the Ve is one of the components partitioned from the total grand variance. If any of the values change in the dataset, all the Ve, Vt, Vs will change correspondingly. In our case discussed in Section 6.1, if we evaluate the real precipitation datasets with the monthly values, the Ve decreases compared to the Ve evaluated by the datasets at the annual scale.

But if we evaluate the method with assumed data as we enlarge the inter-annual variation, Ve is not necessarily decreasing. Ve (or Ue) is an estimation of the difference between multiple datasets. Expanding the temporal variation of a piece of data (or all data) does not mean the difference among multiple datasets decreases or not. It is not very easy to illustrate it in three-dimensional datasets, but we can explain it easily with one-dimensional time series. Assume we have two time series $(T_1, T_2)$ with the same fluctuation $(t_i)$ but different variation amplitudes $(k_1, k_2)$.

$$T_1 = k_1 t_i \; ; \; T_2 = k_2 t_i; \; and \; k_1 < k_2$$

The Ve evaluates the similarity between the two time series. If we increase the variation of $T_1$ (by increasing $k_1$), the similarity of the two series will increase and Ve (or Ue) will decrease. Instead, if we increase the variation of $T_2$ (by increasing $k_2$),, the similarity decreases but Ve (or Ue) increases. So it is not the inter-annual variation change determines the Ve but its difference between the changed data to other series determines the ensemble variance. The conclusion in 1-D series can be upgraded to the 3-D database applied in the manuscript.

However, this decrease of Ve resulted from regular temporal variation instead of variability of ensemble precipitation datasets. In summary, how to separate the influence of normal spatio-temporal variation from the ensemble variability representing the new uncertainty estimation?

Reply: If we only enlarge the temporal variation of a piece of data, the variability of ensemble precipitation datasets will keep the same. But because the data has been partly changed, the difference between the changed dataset and remained others will change. Ve (or Ue) will change as a result.

There are two kinds of uncertainties. The first is the uncertainties between any database and its real values. The second is the uncertainties among multiple datasets which show the different performance of various datasets. The former one is difficult to evaluate because we never know the real values. Thus, we rely on the second uncertainty estimation as if different datasets show small uncertainties around

the ensemble values, the ensemble estimation has high credibility. The present study only focuses on the latter uncertainty which is evaluated by Ve (or Ue). The changes of normal spatio-temporal variation can be revealed in the changes of Ve (or Ue). But the variability of ensemble datasets (the first kind of uncertainty) is not evaluated and it will remain the same if the spatio-temporal variation changes do not affect the ensemble variability.

In addition, is the classical N.t.std or N.s.std affected by the same temporal or spatial variation?

Reply: Taken the sample the reviewer has proposed, if part of the inter-annual variation is enlarged, the N.s.std will keep the same because it requires averaging the time series across different time steps.

$$N.s.std = \frac{\sqrt{\overline{\sigma_{e_t}^2}}}{\mu} = \frac{1}{\mu}\sqrt{\frac{1}{n}\sum_{j=1}^{n}\sigma_{e_t}^2[j]}$$

$$\sigma_{e_t}^2[j] = \sigma^2(\mu_t[j,:])$$

$$\mu_t[j,k] = \frac{1}{m}\sum_{i=1}^{l} z_{ijk}$$

Averaging temporal variation

But N.t.std changes correspondingly. This is the one of the shortcomings of $N.s.std$ (or $N.t.std$) as the changes of temporal variation is not necessarily revealed in the result.

(2) Following Comment (1), the authors should clearly define the reasonable variation resulted from temporal dynamic or spatial heterogeneity and variability associated with uncertainty investigated in the present study of biased ensemble precipitation datasets. Also, as the interaction between the spatio-temporal variation and ensemble data variability exists, is it part of the uncertainty Ve?

Reply: As we explained in the comment (1), there are two kinds of uncertainties and Ve (or Ue) only evaluates one of them. We will explain it better in the revised manuscript.

As the reviewer mentioned in the comment (1), Ve is interacted with Vt and Vs. But we don't think the spatio-temporal variation is interacted with the ensemble data variability. The counter-sample is what the reviewer has suggested in the comment (1) as we enlarged the spatio-temporal variation but the ensemble data variability remains the same. Ve (or Ue) will capture the changes of the spatio-temporal variation of any piece of data but it has no association with the ensemble variability changes.

(3) The authors said the new uncertainty metric Ve contained both temporal and spatial uncertainties at the same time, while the classical metrics (N.t.std or N.s.std) contained only one source of uncertainties. Why the comparison of Ve with classical metrics was conducted by using each of classical metrics rather than the sum of N.t.std and N.s.std?

Reply: Each classical metric has its physical meanings as the $N.s.std$ represents the uncertainties across space and $N.t.std$ represents the uncertainties across time. So, the comparison of Ue to only one of them helps investigate the metric performance on the same physical meaning.

Though it is possible to compare Ue with an combination of $N.s.std$ and $N.t.std$, the two classical metrics cannot be summed up with the same weights due to the different size of their data samples (m and n). Since the Ue is in general linearly correlated with each of the two classical metrics, Ue will be linearly correlated with a new metric which combines the two classical metrics. These can be added into the manuscript after the comparisons of Ue with single metric.

(4) Many literatures in the field of hydro-meteorology have studied on the variance decomposition method in multi-source uncertainty investigation. What's the difference between the new uncertainty estimation partitioned from grand variance and previous studies should be highlighted in Abstract and Introduction.

Reply: We will investigate more literatures and then add the comparisons to the revised manuscript.

(5) There exist many grammar mistakes in the manuscript. For example, "an new uncertainty ..." in Abstract should be "a new uncertainty ...", the expression of "which has been included the model variation" in Abstract is wrong, "of" was omitted after "because" in Line 30 on Page 2. In addition, please check Line 16 on Page 5 and Line 8 on Page 25.

 (6) Despite the grammar mistakes, multiple improper or incomplete English expressions also tended to hinder the readability of the paper. For example, the mean precipitation value in Line 26 on Page 10 may be not only derived from "The long-term annual mean precipitation" but also from the lumping of spatial grids? To make reviewers and readers fully understand this study, the English should be improved considerably throughout the manuscript.

Reply: Thanks for pointing out the mistakes and the improper expressions. We have revised the ones the reviewer has mentioned, and we will check others throughout the manuscript before submitting a new version.

(7) The gauge precipitation provided by CMA was taken as the benchmark. Although the CMA data was excluded from gauge-based group, other gauge-based products also contained part of the gauge data from CMA. This is expected to clearly state. Were the gauge-based data downloaded in grid or gauge format? Are all the precipitation data in daily time scale?

Reply: The CMA was downloaded in grid format and the original data is in daily scale. But we summed up the daily values to monthly and annual scale for use in the present study.

Yes, some of the CMA gauges are included in other global observation systems. We will clarify it in the revised manuscript.

(8) Why is there no content in the section of 2.4?

Reply: Sorry for the mistake. We moved the statements for "underlying of the uncertainties" to previous section. The section 2.4 will be removed in the revised manuscript.

(9) In Figure 6, since the curves plotted represented the uncertainty, what does the band of ±standard deviation around the curves mean, the uncertainty of uncertainty? Please explain.

Reply: The colored solid line represents the ensemble mean of precipitation in each precipitation data group over the same subregion. The shaded area (±standard deviation around the curves mean) represents the uncertainty between different datasets in that data group.

$$\mu_i = \frac{1}{l}(P_{1,i} + P_{2,i} + P_{3,i} + P_{4,i})$$

$$\sigma_i = \sigma(P_{1,i}, P_{2,i}, P_{3,i}, P_{4,i})$$

$P_{1,i}$ represent the mean precipitation in precipitation dataset (the first among the four) over a specific region at time step i. $P_{2,i}, P_{3,i}, P_{4,i}$ represents the values for other three precipitation datasets in the same precipitation groups (gauge-based, merged products, GCMs). The curve is the mean of the four ($\mu_i$) and

the shaded area is their standard deviation ($\sigma_i$). We will revise the caption of Figure 6 to avoid the confusions.

(10) In Figure 12, the quantile of the box is increasing from bottom to top for normal boxplots, while there is inverse order of quantiles here. Please check it.

Reply: Thanks, we did not notice the order here and we will revise it in the new version.

---

## Editor Comment (EC1) · Dimitri Solomatine (Editor) · 17 Aug 2019

The authors addressed the comments adequately, in some cases providing clear answers, but in other cases promising to do more revisions, and to do more work (e.g. on reviewing more literature). It is important to keep these promises and indeed to revise the manuscript accordingly. Please clearly explain what is new in your approach w.r.t. earlier work. This opportunity is now given, and the authors are invited to revise the manuscript. Reviewers (and I) notice quite a number of technical errors and the need to polish English. Please give these aspects your attention.

49, 2019.

---

## Author Response (AR1)

Dear Reviewers,

Thank you so much for your nice comments and kind suggestions! We reply to your questions and comments in this file. In the meantime, we revised the manuscript accordingly and part of the important revision is attached after the replies in this file. The revisions certainly improved the readability and quality of the manuscript compared to its previous version.

RE: Reply to the comments

CR: Corrections in the revised manuscript

Reviewer 1.

General comments The study proposed a new uncertainty estimation that takes into account both spatial, temporal, and model uncertainties. The authors then compared the new uncertainty values with two classic uncertainty metrics and demonstrated the comprehensiveness of the new metric. As the new uncertainty estimation still bears some similarity with the two classical metrics, it could be used as an alternative metric. The reviewer recommends minor revision.

Specific comments Section 2.4 is missing.

    RE: Yes. Instead, we moved the statements for "underlying of the uncertainties" to the previous section.

    CR: The section title has been removed in the revised manuscript.

L16 on page 7: change "Similarity" to "Similarly".
L1 on page 9: change "can also be expressed as the normalized" to "can also be normalized"
L13 on page 9: change "more natural" to "more proper".
L1 on page 10: the use of "global atmospheric gauges" is not proper, change to "global precipitation gauges" instead. Change "representatives" to "representativeness".
L2 on page 10: change "grids dataset" to "gridded dataset". Change "provided by" to "stands for".
L28 on page 10: the percent biases are calculated wrongly. Suppose you use CMA annual precipitation as the base, then the percent biases are: (63.1/589.8)x 100% = 10.7%, and (232/589.8)x 100% =39.3%, respectively.

    RE&CR: We have corrected the above items.

L31-32 on page 10: Do you mean some areas have abrupt precipitation changes rather than following the general gradient? The use of "isolated areas" is confusing to me.

    RE: Thanks for your correction. It is exactly what you're mentioning.

    CR: We corrected the sentence as "For instance, some areas have abrupt precipitation changes rather than following the general tendency. This is probably caused by topography (e.g., the northern Tienshan Mountain, the Qilian Mountains), while it is not captured in the gauge-based products."

L4 on page 12: the description is confusing.

    RE: The sentence was "These differences show the general characteristics and their difference of all the three types of precipitation products. "

The statement is based on Figure 4 which shows the precipitation patterns of three different precipitation groups. Based on the comparisons among the precipitation dataset, the characteristics of each one have been clarified in the words before this sentence.

CR: We revised the sentence as "These differences show the general characteristics of the three types of precipitation products."

L1 on page 14: change "non unit" to "no unit".
L18 on page 14: change "which may has" to "which may have".

RE&CR: We have corrected the above items.

L8-9 on page 16: Figure 6i and 6j do not agree well for gauge-based and merged products, so it is not proper to generalize like this sentence.

RE: The mean values of the merged products are higher than the gauge-based products. But despite of the differences in mean values, the temporal variations (which can be quantified as the correlation), both the gauge-based products and merged products show good correlation with the CMA for all subregions including (i) southwest and (j) northwest China.

CR: The sentences have been modified to: "Despite of the difference in mean values, the temporal evolution of the gauge-based products and merged products agree well with that of the CMA dataset, while the temporal evolution of GCMs ensemble is weaker and not well correlated with that of the CMA."

L15 on page 16: change "divided" to "categorized" or something similar.

RE&CR: We changed "divided" to "categorized" as suggested.

L25-28 on page 16: The comparison between gauge-based products and CMA was mentioned firstly according to Figure 7, and then the reason for the discrepancy between the merged products and CMA was discussed. The transition was missing in between.

RE&CR: Thanks, the transition is added. We added "While the merged products show larger precipitation estimations for most of the regions." Before the explanation of the discrepancy between the merged products and CMA.

L6-12 on page 18: Are the standard deviations of each precipitation data group related to the number of data products that you chose?

RE: We did a test with the GCMs because there are many different available GCMs. When the number of data products increases to a certain number (4-5) the standard deviations (or the variance proportions which come later) will become stable. It is easy to apply the method to more number of products but we limit the used products for 4 because we don't have enough independent gauge-based products or merged products.

L23-33 on page 18: It may be better to denote the subregion numbers in Figure 8, so the audience do not need to go back and forth to identify the subregions.

RE&CR: OK, thanks. All the maps of subregions are numbered, including Figure 8 and 9.

L31-32 on page 20: It seems that higher U_s also correlated to regions with higher model uncertainty in Figure 9 g-i.

RE: Yes. The U_s (the third row in Figure 9) has similar patterns with that of the U_e (the fourth row in Figure 9). This is because in the original datasets, the regions with higher model uncertainty always feature higher spatial heterogeneities (shown in Figure 5). The U_s and U_e proposed in this article further separate and quantify the uncertainty (or heterogeneity) of the two dimensions. However, we focus our discussions on the U_e because of the scope of this study (we state in Line 6-7 on Page 8).

Reviewer 2.

The manuscript entitled "A new uncertainty estimation technique for multiple datasets and its application to various precipitation products" introduced the variance partitioning method into uncertainty quantification of ensemble precipitation datasets, which considers both temporal and spatial uncertainties, and thus established a more comprehensive uncertainty metric as compared with the classic metrics. The deviation of the mathematical framework is rigorous and complete, while lots of work, including various precipitation products in multiple regions, was conducted in the validation of the new metric. On the other hand, some theoretical questions are needed to be explained clear and the English writing of this manuscript needs improvement. The detailed problems are listed as follows:

(1) According to the definition of the new uncertainty metric, it's one of components partitioned from the SST over time, space and ensembles. Thus, the new uncertainty Ve interacted with other components (Vt and Vs), as the authors discussed in Section 6.1. Given an ensemble of precipitation, if we replace one year's data to make the inter-annual variation larger, then the Ve obtained will correspondingly decrease.

RE: Yes, the Ve is one of the components partitioned from the total grand variance. If any of the values change in the dataset, all the Ve, Vt, Vs will change correspondingly. In our case discussed in Section 6.1, if we evaluate the real precipitation datasets with the monthly values, the Ve decreases compared to the Ve evaluated by the datasets at the annual scale.

But if we evaluate the method with assumed data as we enlarge the inter-annual variation, Ve is not necessarily decreasing. Ve (or Ue) is an estimation of the difference between multiple datasets. Expanding the temporal variation of a piece of data (or all data) does not mean the difference among multiple datasets is decreasing or increasing . It is not very easy to illustrate it in three-dimensional datasets, but we can explain it easily with one-dimensional time series.

Assume we have two time series $(T_1, T_2)$ with the same fluctuation $(t_i)$ but different variation amplitudes $(k_1, k_2)$ and $k_1 < k_2$:

$$T_1 = k_1 t_i; T_2 = k_2 t_i; \ \ k_1 < k_2$$

The Ve evaluates the similarity between the two time series. If we increase the variation of $T_1$ (by increasing $k_1$) , the similarity of the two series will increase and Ve (or Ue) will decrease. Instead, if we increase the variation of $T_2$ (by increasing $k_2$),, the similarity decreases and Ve (or Ue) increases. So it is not the inter-annual variation change which determines Ve but its difference between the changed data to other series which determines the ensemble variance. The conclusion in 1-D series can be upgraded to the 3-D database applied in the manuscript.

However, this decrease of Ve resulted from regular temporal variation instead of variability of ensemble precipitation datasets. In summary, how to separate the influence of normal spatio-temporal variation from the ensemble variability representing the new uncertainty estimation?

RE: If we only enlarge the temporal variation of a piece of data, the variability of ensemble precipitation datasets will remain the same. But because the data has been partly changed, the

difference between the changed dataset and remained others will change. Ve (or Ue) will change as a result.

There are two kinds of uncertainties. The first is the uncertainties between any database and its real values. The second is the uncertainties among multiple datasets which show the different performance of various datasets. The former one is difficult to evaluate because we never know the real values. We expect the multiple data sets to converge toward the real value as ensemble means eliminate the random variations associated to measurement errors or chaotic noise for model estimates. Thus, we rely on the second uncertainty estimation as if different datasets show small uncertainties around the ensemble values, the ensemble estimation has a high credibility. The present study focuses on the latter uncertainty which is evaluated by Ve (or Ue). The changes of normal spatio-temporal variation can be revealed in the changes of Ve (or Ue). But the difference between ensemble datasets and the real values (the first kind of uncertainty) is not evaluated and it will remain the same if the spatio-temporal variation changes do not affect the ensemble variability.

CR: We added explanations of the uncertainties we are addressing in this study after the first paragraph of the Introduction:

" As a result, differences exist among various products due to the measurement errors, model biases or chaotic noises. The uncertainty is thus regarded as the deviation of these model results from the their real values.

However, the real values are difficult to measure and uncertainties are difficult to be removed from the datasets. Using ensembles consisting of multiple datasets to generate a weighted average thus becomes very popular in the climate-related researches and the ensemble means are considered as the more reliable estimates."

In addition, is the classical N.t.std or N.s.std affected by the same temporal or spatial variation?

RE: Taken the sample the reviewer has proposed, if part of the inter-annual variation is enlarged, the N.s.std will remain the same value because only the mean value over the entire period is used in the estimates. The enlargement of the variation will not change the mean value, thus the N.s.std remains the same.

$$N.s.std = \frac{\sqrt{\overline{\sigma_{e_t}^2}}}{\mu} = \frac{1}{\mu}\sqrt{\frac{1}{n}\sum_{j=1}^{n}\sigma_{e_t}^2[j]}$$

$$\sigma_{e_t}^2[j] = \sigma^2(\mu_t[j,:])$$

$$\mu_t[j,k] = \frac{1}{m}\sum_{i=1}^{l} z_{ijk}$$

Averaging temporal variation

But N.t.std changes correspondingly. This is the one of the shortcomings of *N.s.std* (or *N.t.std*) as the changes of temporal variation is not necessarily revealed in the result.

CR: The shortcoming of the two classical metrics (N.t.std and N.s.std) is explained in the fourth paragraph of the Introduction. We added "While, the averaging in either dimension means a loss of the information, for instance the data variation. The changes in the variation will not be propagated to the uncertainty estimation if the mean value remains the same." for better understanding the differences between the new variance metric and the two classic metrics.

(2) Following Comment (1), the authors should clearly define the reasonable variation resulted from temporal dynamic or spatial heterogeneity and variability associated with uncertainty investigated in the present study of biased ensemble precipitation datasets. Also, as the interaction between the spatio-temporal variation and ensemble data variability exists, is it part of the uncertainty Ve?

RE: The variation in either the temporal scale or the spatial scale is inherent and each data set provides an specific spatio-temporal pattern. The patterns provided by different datasets vary because of the errors or chaotic noises. The uncertainty measurement in this study (Ve or Ue) is quantifying the differences of the patterns provided by multiple datasets.

When we average the multiple datasets, we obtain the ensemble result and we consider the ensemble data as the believed truth of the measurements. The ensemble data provides another specific spatio-temporal pattern and the spatio-temporal variation of the ensemble data thus has no relation to the Ve any more.

(3) The authors said the new uncertainty metric Ve contained both temporal and spatial uncertainties at the same time, while the classical metrics (N.t.std or N.s.std) contained only one source of uncertainties. Why the comparison of Ve with classical metrics was conducted by using each of classical metrics rather than the sum of N.t.std and N.s.std?

RE: Each classical metric has its physical meanings as the N.s.std represents the uncertainties across space and N.t.std represents the uncertainties across time. So, the comparison of Ue to only one of them helps investigate the metric performance on the same physical meaning.

Though it is possible to compare Ue with an combination of *N.s.std* and *N.t.std*, the two classical metrics cannot be summed up with the same weights due to the different size of their data samples (m and n). The combination could be even more complex. But the combination does not matter for the qualitative comparison because the Ue is in general linearly correlated with each of the two classical metrics, Ue will be linearly correlated with a new metric which combines the two classical metrics.

CR: We added in the manuscript the following paragraph "Each classical metric has its physical meanings as the $N.s.std$ represents the uncertainties across space and $N.t.std$ represents the uncertainties across time. The comparison of $U_e$ to each of them demonstrates the metric performance on the same physical meaning. It is possible to compare $U_e$ with a combination of the two classical metrics, but the combination can be far more complex than a simple sum. However, the qualitative comparison is accessible because $U_e$ has a linear correlation with either of them. The correlation will also remain between $U_e$ and a combination of the two classic metrics by summing up them with certain weights."

(4) Many literatures in the field of hydro-meteorology have studied on the variance decomposition method in multi-source uncertainty investigation. What's the difference between the new uncertainty estimation partitioned from grand variance and previous studies should be highlighted in Abstract and Introduction.

RE: We investigated more literatures and then added the comparisons to the revised manuscript.

CR: We added a paragraph in the introduction. "The total variation is contributed by the uncertainties among different datasets, temporal variation and the spatial heterogeneity. The key to evaluate the uncertainty is to decompose the variation caused by dataset differences from the others. The variation decomposition is often seen in hydro-metrological studies but it is always used to separate the uncertainties generated in each step that propagated to the final variation. For example, Déqué et al. (2007) separated the uncertainties of regional climate models (RCM) to four sources of uncertainties (sampling uncertainty, model uncertainty, radiative uncertainty and boundary uncertainty), and boundary uncertainty plays a greater role. Bosshard et al. (2013)

decomposed the uncertainty in the river streamflow projections to uncertainties from climate models, statistical postprocessing schemes and hydrological models. These implementations differ from the scope of the present study and they fail to separate the uncertainties from the spatio- temporal variations because spatio-temporal averaging has been applied in the estimation process. Sun et al. (2010, 2012) in the first time decomposed the total variation to temporal variation and spatial heterogeneity. However, it is only valid for the one single dataset and thus not able to evaluate the uncertainties if multiple datasets describe the same variable."

(5) There exist many grammar mistakes in the manuscript. For example, "an new uncertainty ..." in Abstract should be "a new uncertainty ...", the expression of "which has been included the model variation" in Abstract is wrong, "of" was omitted after "because" in Line 30 on Page 2. In addition, please check Line 16 on Page 5 and Line 8 on Page 25.

(6) Despite the grammar mistakes, multiple improper or incomplete English expressions also tended to hinder the readability of the paper. For example, the mean precipitation value in Line 26 on Page 10 may be not only derived from "The long-term annual mean precipitation" but also from the lumping of spatial grids? To make reviewers and readers fully understand this study, the English should be improved considerably throughout the manuscript.

   RE&CR: Thanks for pointing out the mistakes (Q5) and the improper expressions (Q6). We have revised the ones the reviewer has mentioned. The manuscript has be refined as well. The tracked corrections can be found in the attached manuscript.

(7) The gauge precipitation provided by CMA was taken as the benchmark. Although the CMA data was excluded from gauge-based group, other gauge-based products also contained part of the gauge data from CMA. This is expected to clearly state. Were the gauge-based data downloaded in grid or gauge format? Are all the precipitation data in daily time scale?

   RE: The CMA dataset uses the densest gauges and probably has the best quality to capture the spatiotemporal variations of the precipitation over the study area. Thus we chose CMA as the benchmark based on expert judgement but we recognize that it is not orthogonal to other gauge products because many of the underlying data is the same. The independence between other gauge-based datasets (e.g., GPCC, CRU) will be stronger.

   The CMA was downloaded in grid format and the original data is in daily scale. The other products are prepared in monthly scale. But we summed up the daily or monthly values to annual scale for use in the present study.

   CR: We added in the methodology part: "All the products of three precipitation types including CMA are in gridded format, though the spatial resolution differs. …. Annual average values are summed up based on their original time steps (daily or monthly) …."

(8) Why is there no content in the section of 2.4?

   RE: Sorry for the mistake. We moved the statements for "underlying of the uncertainties" to previous section and forgot to delete the subsection title.

   CR: The section 2.4 is now removed in the revised manuscript.

(9) In Figure 6, since the curves plotted represented the uncertainty, what does the band of ±standard deviation around the curves mean, the uncertainty of uncertainty? Please explain.

   RE: The colored solid line represents the ensemble mean of precipitation in each precipitation data group over the same subregion. The shaded area (±standard deviation around the curves mean) represents the uncertainty between different datasets in that data group.

$$\mu_i = \frac{1}{l}\left(P_{1,i} + P_{2,i} + P_{3,i} + P_{4,i}\right)$$

$$\sigma_i = \sigma\left(P_{1,i}, P_{2,i}, P_{3,i}, P_{4,i}\right)$$

$P_{1,i}$ represent the mean precipitation in precipitation dataset (the first among the four) over a specific region at time step i. $P_{2,i}, P_{3,i}, P_{4,i}$ represents the values for other three precipitation datasets in the same precipitation groups (gauge-based, merged products, GCMs). The curve is the mean of the four ($\mu_i$) and the shaded area is their standard deviation ($\sigma_i$). We revised the caption of Figure 6 to avoid the confusions.

CR: The caption has been refined and the bold words are added **"…. *the solid curve represents the ensemble mean of precipitation in each precipitation data group over the subregion. The width of shaded area* represents the standard deviation … *The shaded area distributes equally in the two sides of the ensemble mean values for that precipitation group*."**

(10) In Figure 12, the quantile of the box is increasing from bottom to top for normal boxplots, while there is inverse order of quantiles here. Please check it.

RE&CR: Thanks, we did not notice that and we have revised the order in the new version. The descriptions which are relevant to Figure 12 are not affected.

[revised manuscript text omitted]

---

## Referee Report (RR1)

The manuscript entitled "A new uncertainty estimation among multiple datasets and implementation to various precipitation products" was revised based on the reviewer comments. All the comments were responded point by point. The improvement is significant. At the same time, some minor issues still exist, which are listed below.

(1) In Line 15 on Page 15 of the revised manuscript with changes marked, "temporal mean (zone C3)" should be changed to "temporal mean (zone C5)".

(2) The meaning of $i$ dimension in caption of Figure 2 is confusing. Does $i$ represent 1, 2 and 3 for three dimensions?

(3) The "Enseble" in Figure 2 was wrongly spelled.

(4) Is Figure 4 the average annual precipitation over the period of 1979-2005 and over ensemble datasets of a group? Why does the gridded precipitation show the same color in many subregions?

(5) Is the ensemble deviation for the first column in Figure 5 derived by the standard deviation across ensemble precipitation products in a specific group? It should be clearly explained. Similar information should be added in the caption of other figures.

(6) Why do the A1, A2 and A3 in Zone A of Appendix A all have $l$ items? Is it a mistake in spelling?

(7) Please check the spelling in the manuscript carefully.

---

## Editor Decision (ED1)

Please address the comments of the referee.

Additionally: in the new version of manuscript you use the term "derivation ratio". Should it be perhaps the "deviation ratio"?

Comments on the Abstract
================

estimates of a certain climatic variable are frequently seen
-- unclear what do you umean by "seen"... where? By whom?

parallel datasetsd
--- what does parallel mean?

Accompanying uncertainties evaluation with the ensemble is recommended while a fundamental flaw is that the uncertainties in temporal variation and spatial heterogeneity are not together considered for the final uncertainty estimate.
--- cumbersome formulation - please reformulate

Uc is higher than classic estimations metrics for the improvement of uncertainty estimation.
--- Unclear, why it is "for improvement"?

the new uncertainty estimate is more comprehensive than the classic ones as the components are partially identified by the classic metrics.
--- unclear formulation

Multiple precipitation products of different types (gauge-based, merged products and GCMs) are used to better explain and understand the peculiarity of the new methodology
--- unclear how e.g. gauges can explain the new methodology. Consider not using the word "peculiarity"
* * *
The comments above are about the Abstract. I can see similar problems in some other parts of the manuscript. This raises a concern, that the rest of the added and modified text in the manuscript would be also difficult to understand.
Therefore I encourage you to carefully read and revise the text again, giving attention to every sentence. Please also ask help form professional proof-reading services.

---

## Author Response (AR2)

**Replies to the comments from editor and reviewers**

Dear editor and reviewers,

Thank you so much for your comments and suggestions. Based on your comments, we concluded that the presentation quality was the major problem of the pervious submission. Improvement on language was necessary. Therefore, we firstly addressed the comments from editors and reviewers and then revised the manuscript throughout the text to avoid similar problems. We then asked one colleague to read the updated text and revised the manuscript again according to the feedbacks to improve the readability. At the last step, we consulted one UK proofreading service company for the further improvement of the manuscript. A final check was done before submitting the revision. I hope the revised manuscript can meet the publication standard of *HESS* at this time.

The comments have been replied one by one in this document and the manuscript with the changed highlighted has been provided as a separated document. The certificate for proofreading service and the modifications are also attached for your check.
* * *
Editor Comments

Please address the comments of the referee.

> Reply: The concerns of the reviewers have been addressed. We replied to those comments in this document and the corresponding changes can be tracked in the attachments.

Additionally: in the new version of manuscript you use the term "derivation ratio". Should it be perhaps the "deviation ratio"?

> Reply: Sorry for the mistake. It should be "deviation ratio" as you mentioned. This mistake has been corrected in the revised manuscript.

Comments on the Abstract =================

Estimates of a certain climatic variable are frequently seen -- unclear what do you mean by "seen"... where? By whom?

Parallel datasets

--- what does parallel mean?

> Reply: We tried to emphasize that ensemble estimates have been frequently applied in climatic research. The "parallel datasets" indicates a number of datasets which can provide estimates of the same climatic variable. In order to avoid the confusion, we modified the first sentence in the abstract as:
>
> *"Ensemble estimates based on multiple datasets are frequently applied once many datasets are available for the same climatic variable."*

Accompanying uncertainties evaluation with the ensemble is recommended while a fundamental flaw is that the uncertainties in temporal variation and spatial heterogeneity are not together considered for the final uncertainty estimate.

--- cumbersome formulation - please reformulate

> Reply: We included too much information in this single sentence thus we modified the abstract as:

> *"Uncertainty that evaluates the difference between the ensemble datasets is always provided along with the ensemble mean estimates to show to what extent the ensemble members are consistent with each other. However, one fundamental flaw of classic uncertainty estimates is that only the uncertainty in one dimension (either the temporal variability or the spatial heterogeneity) can be considered, whereas the variation along the other dimension is dismissed due to limitations in algorithms for classic uncertainty estimates, resulting in an incomplete assessment of the uncertainties."*

Ue is higher than classic estimations metrics for the improvement of uncertainty estimation. --- Unclear, why it is "for improvement"?

> Reply: The difference of *Ue* and the classic uncertainty metrics is that *Ue* estimate avoids pre-averaging the variation in either the spatial dimension or temporal dimension. This difference results in a larger estimate of *Ue* compared to the classic uncertainty metrics. In the revision, we highlighted the difference in the abstract, but the improvement was explained in detail in the main text.

> *"The new methods avoid pre-averaging in either of the spatiotemporal dimensions and as a result, the Ue estimate is around 20% higher than the classic uncertainty metrics."*

The new uncertainty estimate is more comprehensive than the classic ones as the components are partially identified by the classic metrics.

--- unclear formulation

> Reply: We rewrite this sentence as:

> *"Decomposing the formula for Ue shows that Ue has integrated four different variations across the ensemble dataset members, while only two of the components are represented in the classic uncertainty estimates."*

Multiple precipitation products of different types (gauge-based, merged products and GCMs) are used to better explain and understand the peculiarity of the new methodology --- unclear how e.g. gauges can explain the new methodology. Consider not using the word "peculiarity"

> Reply: We intended to say that the new method is applied to the precipitation products which can be categorized into three groups: gauge-based products, merged products and GCMs. The results of the uncertainty analysis these different precipitation groups help explain the specifics of the new method.

We rewrote the sentence as following.

*"The new approach is implemented and analyzed with multiple precipitation products of different types (e.g., gauge-based products, merged products and GCMs) which contain different sources of uncertainties with different magnitudes. Among the multiple gauge-based precipitation products, Ue is the smallest, while among other products Ue is generally larger because other uncertainty sources are included and the constraints of the observations are not as strong as in gauge-based products."*

The comments above are about the Abstract. I can see similar problems in some other parts of the manuscript. This raises a concern, that the rest of the added and modified text in the manuscript would be also difficult to understand.

Therefore I encourage you to carefully read and revise the text again, giving attention to every sentence. Please also ask help form professional proof-reading services.

Reply: As the editor suggested, we carefully read and revised the manuscript. The modifications have been highlighted in the attachment. We also consulted proofreading service from a UK company, the certificate and the tracked changes can be found in the attachment as well.

Comments from Reviewers

Reviewer 1.

No comments.

Reviewer 2.

The manuscript entitled "A new uncertainty estimation among multiple datasets and implementation to various precipitation products" was revised based on the reviewer comments. All the comments were responded point by point. The improvement is significant. At the same time, some minor issues still exist, which are listed below.

1. In Line 15 on Page 15 of the revised manuscript with changes marked, "temporal mean (zone C3)" should be changed to "temporal mean (zone C5)".

   Reply: Yes. That should be zone C5 and we have corrected this mistake in the updated manuscript.

2. The meaning of i dimension in caption of Figure 2 is confusing. Does i represent 1, 2 and 3 for three dimensions?

   Reply: Yes, we intended to use $i$ to represent the three dimensions in Figure 2. But since $i$ has been used as the index for the temporal dimension in equations, we alternatively use $x,y,z$ to represent one of the three dimensions ($s,t,e$) in Figure 2. The changes are highlighted in the caption:

*"Partitioning the temporal-spatial-ensemble variance. The original database is re-organized into three dimensions: time, space and ensemble. Zones with different colours represent different processes based on the original database through different dimensions. The labels of the zones are listed on the right; detailed definitions can be found in Appendix A. The grand variance is $\sigma^2$ and the grand mean is $\mu$. The subscripts $t$, $s$, and $e$ indicate dimensions of time, space and ensemble, respectively. In Zone A, $\mu_x$ shows the mean values across the $x$-dimension ($x$=$t$, $s$ or $e$); in Zone B, $\sigma^2\_x$ indicates the variation across the $x$-dimension; in Zone C, $\sigma_{x\_y}^2$ indicates the variation across the $x$-dimension of $\mu_y$ ($y$=$t$, $s$ or $e$); in Zone D $\mu_{xy}$ indicates the means across the $x$- and $y$-dimensions; in Zone E, $\sigma_{xy}^2$ indicates the variation across the $x$- and $y$-dimensions; in Zone F, $\sigma_x^2(\mu_{yz})$ indicates the variation across the $x$-dimension of the means across the $y$- and $z$-dimensions ($z$=$t$, $s$ or $e$)."*

3. The "Enseble" in Figure 2 was wrongly spelled.

   Reply: It has been corrected in the updated figure.

4. Is Figure 4 the average annual precipitation over the period of 1979-2005 and over ensemble datasets of a group? Why does the gridded precipitation show the same color in many subregions?

   Reply: Yes, Figure 4 shows the average annual precipitation over 1979-2005 in grids for each group of the precipitation products (the caption has been revised for less confusion). We use a discrete intervals color bar rather than a continuous color bar in this graphic to reduce the difficulty of capturing the spatial patterns of the precipitation. In this case the grids in a same precipitation interval (400 mm/yr) are marked in the same color. We chose 400mm/yr because it is one of the criterions that distinguishes the climate types (e.g., dry area with precipitation less than 400mm/yr, semi-dry and semi-wet area with precipitation among 400-800mm/yr, wet area with precipitation larger than 800mm/yr). So, we can simply tell the climatic types from the precipitation using this discrete interval color bar.

5. Is the ensemble deviation for the first column in Figure 5 derived by the standard deviation across ensemble precipitation products in a specific group? It should be clearly explained. Similar information should be added in the caption of other figures.

   Reply: Yes, the standard deviation is estimated among the precipitation products within a specific group. For instance, the first row shows the results for gauge-based products and the second row shows the results for merged products and the third row is for GCMs.

   We have revised the captions for Figure 5:

   *"The spatial distribution of model uncertainty in annual precipitation among different ensemble products. The uncertainty is expressed as the standard deviation of the annual precipitation across ensemble precipitation products of a specific group (up: gauge-based products, middle: merged products, bottom: GCMs). The left panels are the values of the uncertainty. The right panels are the ratios of ensemble deviation to the ensemble means of the datasets in the corresponding group."*

The captions from Figure 6 to Figure 11 are also revised if necessary.

6. Why do the A1, A2 and A3 in Zone A of Appendix A all have l items? Is it a mistake in spelling?

Reply: Sorry that it is our mistake. The upper bound should be *m, n, l* corresponding to *t, s* and *e* dimensions. The errors have been corrected in the Zone A and in the Zone B as well. A screenshot for the modification is attached for your check.
* * *
**Appendix A: The algorithms for different expressions in the methodology**

Zone A:

A1: $\mu_t[s,e;n \times l]; \mu_t[j,k] = \frac{1}{m}\sum_{i=1}^{l} z_{ijk} \mu_t[s,e;n \times l]; \mu_t[j,k] = \frac{1}{m}\sum_{i=1}^{m} z_{ijk}$

A2: $\mu_s[e,t;l \times m]; \mu_s[k,i] = \frac{1}{n}\sum_{j=1}^{l} z_{ijk} \mu_s[e,t;l \times m]; \mu_s[k,i] = \frac{1}{n}\sum_{j=1}^{n} z_{ijk}$

A3: $\mu_e[t,s;m \times n]; \mu_e[i,j] = \frac{1}{l}\sum_{k=1}^{l} z_{ijk}$

   Zone B:

B1: $\sigma_t^2[s,e;n \times l]; \sigma_t^2[j,k] = \frac{1}{m}\sum_{i=1}^{l}(z_{ijk} - \mu_t[j,k])^2 \sigma_t^2[s,e;n \times l]; \sigma_t^2[j,k] = \frac{1}{m}\sum_{i=1}^{m}(z_{ijk} - \mu_t[j,k])^2$

B2: $\sigma_s^2[e,t;l \times m]; \sigma_s^2[k,i] = \frac{1}{n}\sum_{j=1}^{l}(z_{ijk} - \mu_s[k,i])^2 \sigma_s^2[e,t;l \times m]; \sigma_s^2[k,i] = \frac{1}{n}\sum_{j=1}^{n}(z_{ijk} - \mu_s[k,i])^2$

B3: $\sigma_e^2[t,s;m \times n]; \sigma_e^2[i,j] = \frac{1}{l}\sum_{k=1}^{l}(z_{ijk} - \mu_e[i,j])^2$
* * *
7. Please check the spelling in the manuscript carefully.

Reply: Following your suggestions, we carefully revised the manuscript and consulted professional proofreading service to avoid the spelling errors in the manuscript.

Manuscript with changes highlighted.

[revised manuscript text omitted]

Certificate of proofreading service & highlighted changes

[Figure]

Proof-Reading-Service.com Ltd, Devonshire
Business Centre, Works Road, Letchworth Garden
City, Hertfordshire, SG6 1GJ, United Kingdom
Office phone: +44(0)20 31 500 431
E-mail: enquiries@proof-reading-service.com
Internet: http://www.proof-reading-service.com
VAT registration number: 911 4788 21
Company registration number: 8391405

10 January 2020

To whom it may concern,

**RE: Proof-Reading-Service.com Editorial Certification**

This is to confirm that the document described below has been submitted to Proof-Reading-Service.com for editing and proofreading.

We certify that the editor has corrected the document, ensured consistency of the spelling, grammar and punctuation, and checked the format of the sub-headings, bibliographical references, tables, figures etc. The editor has further checked that the document is formatted according to the style guide supplied by the author. If no style guide was supplied, the editor has corrected the references in accordance with the style that appeared to be prevalent in the document and imposed internal consistency, at least, on the format.

It is up to the author to accept, reject or respond to any changes, corrections, suggestions and recommendations made by the editor. This often involves the need to add or complete bibliographical references and respond to any comments made by the editor, in particular regarding clarification of the text or the need for further information or explanation.

We are one of the largest proofreading and editing services worldwide for research documents, covering all academic areas including Engineering, Medicine, Physical and Biological Sciences, Social Sciences, Economics, Law, Management and the Humanities. All our editors are native English speakers and educated at least to Master's degree level (many hold a PhD) with extensive university and scientific editorial experience.

[revised manuscript text omitted]

The two uncertainty estimates (Eqs  28 and 29) correspond to the two classic metrics presented in the Introduction. We will compare  $U_e$ with these two classic metrics ($N.t.std$ and $N.s.std$) to show their relations and differences.

**2.3 Study area and data description**

 Mainland China has been selected as the study area because of its large area and different  types of climate (**?**). Ten different subregions  have been defined to facilitate the comparisons and analysis  of the strong spatial variations. The subregions are  (1) Songhua River Basin, (2) Liao River Basin, (3) Hai River Basin, (4) Yellow River Basin, (5) Huai River Basin, (6) Yangtze River Basin, (7) Southeast China, (8) South China, (9) Southwest China, (10) Northwest China , see Figure 3. **Author query: There are several issues with 'the southern China', which is certainly incorrect. The important issue is that one cannot write 'the southern China' although one can write 'the southern part of China' or 'southern China' without the 'the'. The less important issue is that southern China necessarily includes southeastern China and southwestern China. More usual, therefore, is 'South China', 'Southwest China', etc., where the meaning is whatever is generally understood, but South China does not have to logically include Southwest China unless that is what is generally understood. For example, in the U.S., 'the South' does not include 'the Southwest', for traditional and historical reasons, whereas southern U.S. does necessarily include the Southwest. Now, this issue is less important, and you can go back to what you were writing provided you omit your 'the' (or include 'part of'). The reason I have made this change consistently is because it would be easier for you to revert these changes than to implement my suggestions on your own.** 
[revised manuscript text omitted]

---

## Author Response (AR3)

**Reply to the comments**

Dear editor and reviewer,

Thank you again for your comments and suggestions. We revised the manuscript based on all the comments and we checked throughout the text for all possible improvements. The manuscript has been reviewed by the co-authors and further updated by their comments.

The editor and reviewer's comments have been replied one by one in this document and the manuscript with the changed highlighted has been provided as a separated document.
* * *
Editor's comments:

Comments to the Author:
We are one step before the publication.
Still, after two rounds, there are comments of the referee - please address them carefully.
Additionally: there are still pieces of text which are not formulated well...
E.g. on line 2 of the Abstract:
"Uncertainty that evaluates the difference".
Uncertainty cannot evaluate the difference. Perhaps you wanted to say this:
"Uncertainty metrics based on the difference".

Or in Conclusion:
"Using the classic metrics, in most regions, the GCMs have been indicated as having the largest model uncertainty."
I think it is better to formulate this as follows:
"Using the classic metrics, in most regions, uncertainty of GCMs have been found to be the largest."

Please check the text again - you may find more examples like this. Perhaps you could ask the second author, with a lot experience in publishing, to carefully read the manuscript...

RE: We refined the above two sentences.

We checked and avoided using "that" in similar cases. We found the latter expression mainly occurs when we describe the uncertainty values over different subregions. Accordingly, we refined all those sentences as suggested.
* * *
Comments from reviewers:

I have no other comments except for several small queries, which are listed as follows.

(1)     "A new uncertainty estimation" in the title is used uncommonly. Is it necessary to express it as "A new uncertainty estimation method/technique"?

RE: We modified the title as "a new uncertainty estimation **approach**" as suggested.

(2)     Do the authors mean "underestimate" when using "reduce" in "This shows that classic metrics reduce the uncertainty estimate"?

RE: Yes. It should be "underestimate" in this case.

(3)     In the expression of "Among the multiple gauge-based precipitation products, Ue is the smallest", is "among" appropriate? I think Ue is derived from a specified group of ensemble precipitation datasets rather than is a gauge-based precipitation product. Similar expression can be found in other places throughout the manuscript.

RE: Thanks. We modified the similar phases in the manuscript as "uncertainty **of** the products in a specific group" and "differences/similarities **among** the products in a specific group". The modifications can be easily tracked in the attached file with all changes highlighted.

Regarding the mentioned sentence, Instead of using "*Among* the multiple gauge-based precipitation products, Ue is the smallest", we rephased it as "Ue of the gauge-based precipitation products is the smallest, …".

(4)     In the second line of Section 2.1, "of" is suggested to add between "dimensions" and "(1)".

RE: We added "of" in section 2.1 as suggested.
Similarly, in the section 4.1, when we describe the size of the used data, we use a colon (:) which is more appropriate than "of" in this case.

*"As introduced in the methods section, the input annual precipitation to the approach is re-organized into three dimensions: (1) **time**, 27 years from 1979 to 2005, (2) **space**, 0.5° grids in a specific region and (3) **ensemble**, the number of models in each precipitation group (four models for each of the three groups)."*

(5)     In the explanation of Equ. (4), i.e., "The total variation receive contributions …", please change "receive" into "receives".

RE: We corrected this error.

(6)     Please remove "the" at the beginning of the captions of figures, if any. For example, "the" in "The annual precipitation" in Figure 4.

RE: We modified the captions of Figures 1, 4, 5, 6, 7 and 10 as suggested. Caption of Table 1 is modified as well.

(7) Finally, please carefully check the accuracy of prepositions and the form of inflective verbs used in the manuscript.

RE: We have checked the manuscript and revised all inappropriate prepositions and errors in forms of inflective verbs that can be found.

[revised manuscript text omitted]